# Mechanism of sensitivity modulation in the calcium-sensing receptor via electrostatic tuning

Michael R. Schamber [1] & Reza Vafabakhsh [1✉]

Transfer of information across membranes is fundamental to the function of all organisms and is primarily initiated by transmembrane receptors. For many receptors, how ligand sensitivity is fine-tuned and how disease associated mutations modulate receptor conformation to allosterically affect receptor sensitivity are unknown. Here we map the activation of the calcium-sensing receptor (CaSR) - a dimeric class C G protein-coupled receptor (GPCR) and responsible for maintaining extracellular calcium in vertebrates. We show that CaSR undergoes unique conformational rearrangements compared to other class C GPCRs owing to specific structural features. Moreover, by analyzing disease associated mutations, we uncover a large permissiveness in the architecture of the extracellular domain of CaSR, with dynamics- and not specific receptor topology- determining the effect of a mutation. We show a structural hub at the dimer interface allosterically controls CaSR activation via focused electrostatic repulsion. Changes in the surface charge distribution of this hub, which is highly variable between organisms, finely tune CaSR sensitivity. This is potentially a general tuning mechanism for other dimeric receptors.

[1] Department of Molecular Biosciences, Northwestern University, Evanston, IL 60208, USA. ✉email: reza.vafabakhsh@northwestern.edu

Every cell must translate extracellular information from chemical or physical signals to the biochemical language of the cell. This task is usually initiated by membrane receptors and often requires allosteric communication between the extracellular ligand-binding region and the intracellular effector region of the receptor. Receptor dimerization upon ligand binding is a ubiquitous mechanism to initiate signal transduction and is widely used in receptors, such as bacterial chemotaxis receptors, receptor tyrosine kinases (RTKs), cytokine receptors, and some GPCRs[1–3]. This approach is generalizable and has been used to engineer synthetic switches or modulate signaling output of native receptors by artificial ligands[1,4,5]. However, a mechanistic understanding of how different ligands tune signaling output after dimerization and the relationship between variations in receptor topology and signal strength are lacking for many receptors.

GPCRs are the largest family of allosteric membrane receptors in human. Among them, class C GPCRs are constitutive dimers with ~600 amino acid extracellular domain that, in some members, are covalently linked[6,7] (Fig. 1a). Members of class C GPCRs include mGluRs, GABABRs, CaSR, and sweet and umami taste receptors, and are activated by L-amino acids and ions[8–11]. The canonical ligand recognition and binding site in class C GPCRs is within the conserved bilobed enus flytrap (VFT) domain[10–14], which is evolutionarily related to bacterial periplasmic amino acid-binding proteins (PBPs), such as the leucine-binding protein and the leucine/isoleucine/valine-binding protein[15–18]. Crystal structures of CaSR extracellular domain (ECD) have provided insights into the ligand binding pockets and structural rearrangements upon agonist binding[12,13]. More recently, cryo-electron microscopy (cryo-EM)

has provided insights into how conformational changes are propagated to the seven-transmembrane-helix domain (7TM) in the full-length receptor and how various calcilytic or calcimimetic compounds target CaSR[14–18].

Although mGluRs and CaSR have a similar domain architecture, high structural homology (RMSD = 1.7 Å) and sense similar types of ligands, they have evolved to sense signals with very different temporal profiles. In the case of mGluRs, which are primarily synaptic receptors, glutamate exists at very low basal levels which increases to millimolar concentration upon release from synaptic terminals and on a very fast time scale (10 ms), analogous to a digital signal[19,20]. However, in the case of CaSR, serum $Ca^{2+}$ varies continuously around 2.4 mM on a slow time scale (hours)[21], like an analog signal. How the structure and dynamics of the two receptors have evolved to match the temporal characteristic of their signal is unclear.

In all vertebrates from fish to humans, CaSR functions as the molecular sensor for the extracellular calcium concentration[22–24]. In this context, CaSR provides corrective feedback to keep serum calcium levels within a very narrow homeostatically controlled range (2.2–2.6 mM in human)[25]. However, the $EC_{50}$ of CaSR varies widely between different animals, likely to match the $Ca^{2+}$ homeostatic requirement and environmental niche of each organism. For example, published $EC_{50}$ values range from 1 mM in goldfish to 7.5 mM in dogfish shark[26,27]. How evolution tunes the sensitivity of CaSR is unclear.

Finally, numerous disease-associated mutations in CaSR have been identified[28], which generally cause hypercalcaemic or hypocalcaemic disorders as well as some cancers[29]. Many of these mutations are away from known ligand-binding sites on CaSR

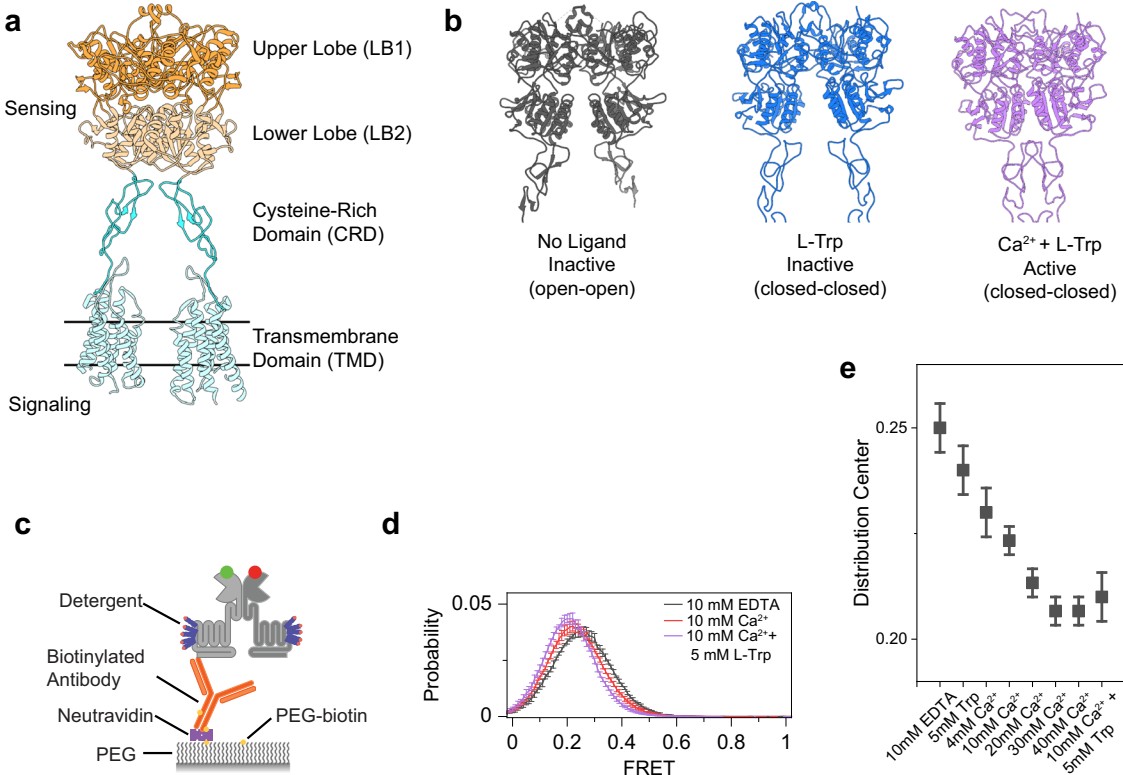

**Fig. 1 CaSR is an intrinsically dynamic receptor, and ligands stabilize the active conformation. a** Ribbon representation the CaSR structure (PDB ID: 7DTW) colored to highlight the upper lobe (orange), lower lobe (pale orange), cysteine-rich domain (cyan), and the transmembrane domain (pale cyan). **b** Ribbon representation of the CaSR ectodomain in the Ioo, Icc, and Acc conformations (PDB IDs: 5K5T, 7DTU, and 7DTV respectively). **c** Schematic of single-molecule FRET experiments. **d** smFRET population histograms in the presence of 10 mM EDTA, 10 mM $Ca^{2+}$, or 10 mM $Ca^{2+}$ and 5 mM L-Trp. Data represent mean ± s.e.m. of $n = 3$ independent biological replicates. **e** Center of a single gaussian distribution fit to FRET histograms. Data represents the mean ± s.e.m. of $n = 3$ fits to independent biological replicates.

and exert their effect allosterically by modulating the intersubunit cooperativity and crosstalk. A general framework for understanding how these mutations alter receptor topology and dynamics to influence the strength of signal output is unknown.

To study these questions, we use sequence analysis, signaling assays, and single-molecule FRET (smFRET) imaging and map the conformational dynamics of the extracellular domain of CaSR in the presence of different CaSR ligands. We use a site-specific labeling method based on unnatural amino acid incorporation to avoid local structure disruption caused by conventional labeling strategies. We find that the extracellular domain of CaSR is dynamic, and the receptor continuously samples the active state in the absence of ligands. We show that two unique structural features in CaSR contribute to the unique conformational dynamics of CaSR compared with other class C GPCRs. Specifically, we identify a role for a focused negatively charged patch at the dimer interface of CaSR that limits the receptor residency in the active state via electrostatic repulsion, and we demonstrate an interprotomer loop restricts the conformation space of the VFT of CaSR and reduces the effect of amino acid binding on $Ca^{2+}$ sensitivity. We propose that this design is likely a general mechanism for fine-tuning sensitivity in many dimeric receptors and a plausible strategy for therapies.

## Results

### Mapping the conformational dynamics of CaSR N-terminal domain.
Structural and spectroscopic studies[9–13,30–39] suggest a universal activation mechanism for class C GPCRs where ligand binding in the VFT domain results in a conformational rearrangement in the ECDs which propagates over 10 nm to rotate and bring the 7-transmembrane (7TM) domains closer and activate the receptor[11,30,36,40]. Structures of CaSR have shown different conformations with different degrees of rearrangement in various ligand states and generally implies an activation mechanism for CaSR similar to mGluRs[12,14,15] that consists of three conformational states: the inactive open–open (Ioo) conformation characterized by an open VFT and separation between LB2–LB2 and CRD–CRD interfaces; the inactive closed–closed (Icc) conformation characterized by a closed VFT and separation between LB2–LB2 and CRD–CRD interfaces; and the active closed–closed (Acc) conformation characterized by VFT closure and engagement of the LB2–LB2 and CRD–CRD interfaces (Fig. 1b). However, a direct demonstration of the conformational dynamics underlying CaSR activation and how conformational changes between different domains of CaSR are coupled is lacking.

We first focused on resolving the ligand-independent and ligand-dependent conformational changes at the N-terminal domain of CaSR. Atomic structures of CaSR in the presence and absence of agonists suggest only a small movement of the N-terminal domain of CaSR upon activation[12–15] in contrast to mGluRs[10,30,39]. However, recent results from bulk time-resolved FRET measurements suggest ECD of CaSR undergoes large conformational rearrangement similar in the magnitude to the change in mGluRs[36]. Because receptors undergo rapid and unsynchronized transitions between multiple conformational states, it is inherently challenging to convert bulk FRET measurements to distances and map their activation process and kinetics. To overcome these limitations, we performed single-molecule Förster resonance energy transfer (smFRET) experiments to resolve the ligand-independent and ligand-induced conformational dynamics of CaSR.

First, we performed experiments on a CaSR construct with an N-terminal SNAP-tag for fluorescent labeling and a C-terminal FLAG-tag for surface immobilization (Fig. 1c). This construct allows direct comparison with similarly tagged mGluR2 construct[39]. Functional experiments using calcium imaging in cells showed that this construct is active with an $EC_{50}$ of 2.84 mM (Supplementary Fig. 1a, b), comparable to the wildtype receptor[22,41]. In single-molecule experiments, the FRET distribution showed a single peak centered at FRET = 0.25 in the absence of CaSR ligands (and in the presence of 10 mM EDTA) (Fig. 1d) corresponding to the Ioo conformation. The activating condition (10 mM $Ca^{2+}$ and 5 mM L-Trp) shifts the FRET distribution to 0.21 FRET (~2.5 Å distance change) (Fig. 1d) corresponding to the Acc conformation. This FRET change is different from what was observed with a similarly SNAP-tagged mGluR2 that showed a large FRET shift from 0.45 to 0.2 (~12.2 Å) upon activation[39]. Therefore, this result is consistent with the smaller rearrangement of the VFT domains of CaSR observed in atomic structures and supports a unique activation mechanism for CaSR compared to mGluRs. Quantification of receptor dynamics by cross-correlation analysis showed that the receptor is most dynamic in the absence of any ligand compared to conditions with ligands as quantified by cross-correlation amplitude (Supplementary Fig. 1c, Supplementary Data 1). This is again in contrast with mGluR2 which was not very dynamic in the apo and fully active states[39]. The observation that CaSR frequently and very briefly visits the 0.21 FRET state at room temperature is consistent with the conformational heterogeneity observed for the ligand-free cryo-EM structures of CaSR[14]. In the presence of agonists, the amplitude of the cross-correlation was reduced (Supplementary Fig. 1c) and the FRET distribution narrowed (Supplementary Fig. 1d), suggesting that agonists stabilize VFT dynamics. Calcium alone was able to fully shift the FRET distribution to the FRET state corresponding to the Acc conformation (Fig. 1e). To better visualize receptor dynamics, we acquired data at 5 ms and observed FRET histograms consistent with our 30 and 100 ms data (Supplementary Fig. 1e,f). At this increased time resolution, single-molecule traces were still very dynamic (Supplementary Fig. 1g) and, in the absence of ligand, showed very brief transitions, many within one time point (Supplementary Fig. 1g). This again is in contrast with mGluRs that are stable in the inactive state in the absence of ligand and show long-lived visits to the active state in the presence of agonists[39]. Together, these results confirm that in the absence of the ligand, CaSR visits the active state frequently, suggesting a low energy barrier between the active and inactive states in CaSR and in contrast with the mGluRs. However, these brief and unproductive visits do not result in receptor activation. On the other hand, $Ca^{2+}$ stabilizes the active state conformation to activate the receptor.

Because of the orientation of SNAP-tags in CaSR, the FRET change between the active and inactive FRET states is small (ΔFRET = 0.04). This limited our ability to further quantify the kinetic details of the activation process. To overcome this limitation, we designed a new smFRET sensor, based on unnatural amino acid (UAA) 4-azido-L-phenylalanine incorporation[40,42–44] at the residue 451 (D451UAA hereafter), which expressed at high levels and labeled very efficiently[45,46] (Fig. 2a). The residue 451 is at the top surface of the LB1 in CaSR with 51.5 Å distance between monomers (Fig. 2b). This distance put donor and acceptor probes in the sensitive range for FRET and provide higher spatial resolution than the N-terminal SNAP construct. First, we verified that D451UAA showed similar conformational dynamics as the SNAP-tag construct. Both the inactive (3 mM EDTA) and fully active (10 mM $Ca^{2+}$ + 5 mM L-Trp) conditions showed a single FRET distribution peak centered on 0.41 and 0.29, respectively (Fig. 2c), corresponding to a FRET change of 0.12 or an approximate 4.8 Å change in distance. This is again consistent with our previous result of a small VFT domain conformational rearrangement upon activation and

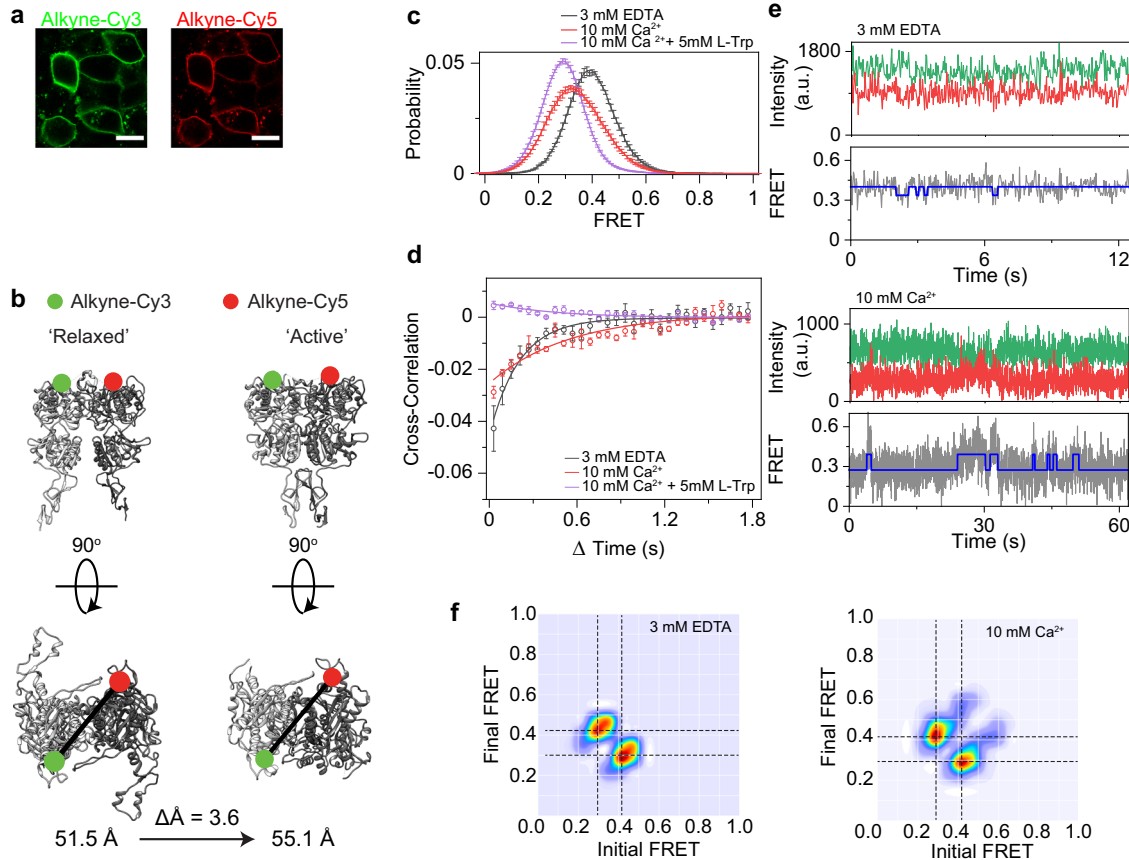

**Fig. 2 D451UAA CaSR identifies transitions between two conformational states. a** Representative confocal image of HEK293T cells expressing D451UAA and labeled with Alkyne-Cy3 (left) and Alkyne-Cy5 (right). Scale bar, 15 µm. **b** Ribbon representation of crystal structures PDB ID: 5K5T (left) and PDB ID: 7DTV (right) showing approximate location of D451UAA labeling. Measured distance is between D451 Cα represented by black line. **c** smFRET population histogram in the presence of 3 mM EDTA, 10 mM Ca$^{2+}$, or 10 mM Ca$^{2+}$ and 5 mM L-Trp. Data represent mean ± s.e.m. of $n = 3$ independent biological replicates. **d** Cross-correlation plots in the presence of 3 mM EDTA, 10 mM Ca$^{2+}$, or 10 mM Ca$^{2+}$ + 5 mM L-Trp. Data represent ±s.e.m. of $n = 3$ independent biological replicates. Data was fit to a single exponential decay function. **e** Sample single molecule traces of D451UAA in 3 mM EDTA (top) and 10 mM Ca$^{2+}$ (bottom) showing donor (green) and acceptor (red) intensities, corresponding FRET values (gray), and idealized FRET trajectory from HMM fit (blue). **f** Transition density plot of D451UAA. Dashed lines represent the most frequently observed transitions and were used for multiple-peak fitting of FRET histograms.

published structures that show 3.6 Å difference between the Ioo and Acc conformations (Fig. 2b)[14,15]. Like the N-terminal SNAP sensor, addition of ligands reduced the cross-correlation amplitude of D451UAA (Fig. 2d). Single-molecule traces showed brief visits to the active state (Fig. 2e). Addition of Ca$^{2+}$ increased the relative occupancy of the active state (Fig. 2c). Analysis of single-molecule traces using a Hidden Markov model (HMM) also verified that ligand-independent and Ca$^{2+}$-dependent transitions are between the inactive (FRET = 0.41) and active (FRET = 0.29) states (Fig. 2e, f and Supplementary Fig. 2a). Together, these results show that the D451UAA sensor provides a higher sensitivity to conformational change and confirms the results from the SNAP sensor.

**Amino acids facilitate VFT rearrangement beyond VFT closure.** In mGluRs and GABA$_B$Rs, binding of glutamate or GABA within the VFT domain ligand-binding site is sufficient for receptor activation. In CaSR, the canonical class C GPCR ligand binding site is promiscuous and binds different L-amino acids[8]. However, whether L-amino acid binding alone is sufficient or necessary for CaSR activation has been a matter of debate[8,12,14,36,41,47]. Moreover, while evidence suggests L-amino acids act as positive allosteric modulators (PAMs)[48], the dynamic

mechanism of this modulatory effect is not well understood. We used the D451UAA sensor to investigate the effect of L-Trp on CaSR structure and dynamics. Published cryoEM structures show a distance change of −0.7 Å measured at the D451 Cα as the receptor transitions from the Ioo conformation to the Icc conformation upon L-Trp binding (Fig. 3a). This change is below the resolution of smFRET and therefore these two states should appear as a single peak in our smFRET measurement. However, we observed that 2.5 mM L-Trp alone resulted in a smFRET peak centered between the active and inactive FRET states (Fig. 3b). This implies that amino acid binding is causing a change in conformation beyond that shown in the Icc structure. The unexpected change in FRET could be caused by the stabilization of a novel conformation or induction of rapid exchange between the 0.29 and 0.41 FRET states due to L-Trp binding. To test this, we performed an L-Trp titration, and we found that by increasing L-Trp concentration the FRET distribution peak moved towards lower FRET values (Fig. 3b), consistent with L-Trp increasing transitions between the active and inactive states. Inspection of single-molecule traces in the presence of L-Trp alone also showed that CaSR can briefly visit the 0.29 FRET state (Fig. 3c). HMM analysis also verified that in the presence of L-Trp, CaSR transitions between active and inactive states (FRET = 0.41 and FRET = 0.29) (Fig. 3d). However, like the unliganded receptor,

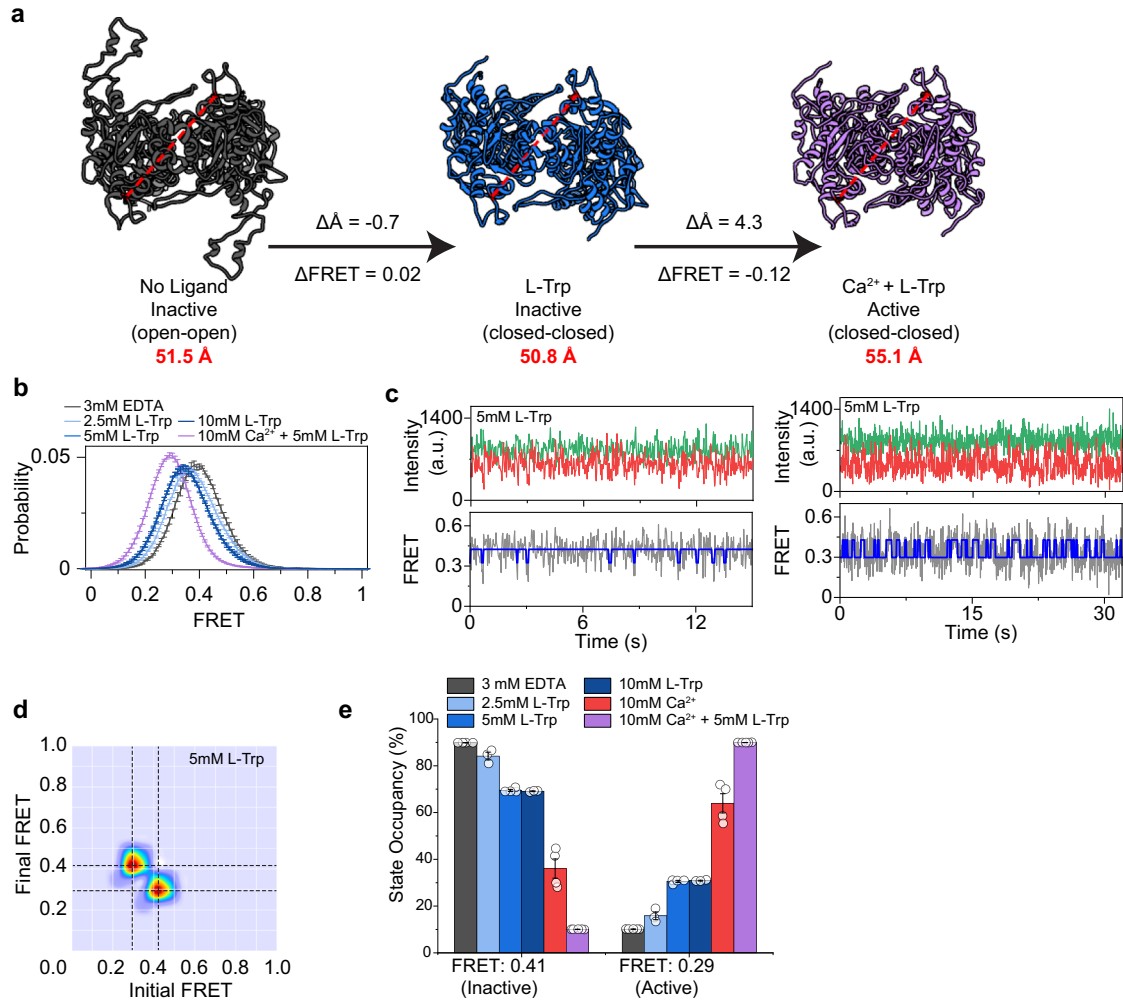

**Fig. 3 L-Trp increases occupancy of the active state but is insufficient for activation. a** Top-down view of CaSR structures showing the distance between the Cα of D451 (red) for the Ioo, Icc, and Acc conformations (PDB IDs: 5K5T, 7DTU, and 7DTV, respectively). Arrows show change in distance of the D451 Cα and the corresponding predicted change in FRET. **b** smFRET population histogram in the presence of 3 mM EDTA, 2.5 mM L-Trp, 5 mM L-Trp, 10 mM L-Trp, or 10 mM $Ca^{2+}$ and 5 mM L-Trp. Histograms for 5 mM L-Trp and 10 mM L-Trp overlap. Data represent mean ± s.e.m. of $n = 3$ individual independent biological replicates. **c** Sample single molecule traces of D451UAA in 5 mM L-Trp showing donor (green) and acceptor (red) intensities, corresponding FRET (gray), and idealized FRET trajectory from HMM fit (blue). Sample traces show particles exhibiting different behaviors in the same condition with infrequent and very brief transitions (1–2 datapoints), or frequent and brief transitions (5–10 data points). **d** Transition density plot of D451UAA. Dashed lines represent the most frequently observed transitions and were used for multiple-peak fitting of FRET histograms. **e** Occupancy of the two FRET states of the VFT in the presence of increasing ligand concentrations. Values represent mean ± s.e.m. area under individual FRET peaks from $n = 3$, 4, or 5 independent biological replicates.

the transitions are brief and do not substantially increase the active state occupancy and likely cannot result in signaling[14,36,41]. Therefore, these results are consistent with a model that amino acids not only induce closure of the VFT as observed in structures, but also increase the occupancy of the 0.29 FRET state suggesting transient rearrangement of the dimeric interface.

To investigate the cooperative function of L-Trp with $Ca^{2+}$, we quantified the occupancy of each FRET state in different ligand concentrations (Fig. 3e and Supplementary Fig. 2b). We found that while 10 mM $Ca^{2+}$ alone significantly increases the occupancy of the active state, addition of 5 mM L-Trp fully shifts the occupancy of the active state. Importantly, higher concentrations of $Ca^{2+}$ alone can fully shift the histogram to the lower FRET peak (Fig. 1e). To ensure that our observation is not due to residual L-Trp bound to CaSR during the purification[12–14,16], we performed an extended wash of immobilized receptor and did not detect any change in the FRET distribution post-wash while the distribution again shifted to lower FRET in the presence of $Ca^{2+}$

alone (Supplementary Fig. 2c). Furthermore, we collected smFRET data before, during, and after addition of 5 mM L-Trp. We found that 5 mM L-Trp reversibly shifted the FRET histogram (Supplementary Fig. 2d) suggesting we can remove amino acids bound to immobilized receptors. Based on these data, it is unlikely that the effect observed in the presence $Ca^{2+}$ alone to be caused by residual amino acid binding. Together, these observations are consistent with the role of amino acids as allosteric modulators[14,36].

Next, we probed the propagation of conformational changes beyond the VFT. According to the canonical class C GPCR activation model, the extracellular domain controls the proximity and the relative orientation of the 7TM domains, and in the active state, the 7TM domains come into direct contact[14,15,30,33,40]. We used another UAA-based FRET sensor at amino acid 593 (E593UAA) to probe the conformation of the CRD of CaSR as it showed the highest labeling efficiency of CRD sensors tested (Supplementary Fig. 3a) and had an

activity comparable to wild-type (Supplementary Fig. 3b). Since the CRD is closer to the 7-TM domain we expected that this sensor to be a more accurate reporter of receptor activation than the VFT domain sensor. SmFRET analysis showed that the CRD of CaSR is in dynamic equilibrium between at least four conformational states (Fig. 4a, b), similar to mGluR2[40], and Ca$^{2+}$ increases the occupancy of the higher FRET states consistent with the compaction of the receptor upon activation (Fig. 4c and Supplementary Fig. 3c). Importantly, we found that L-Trp alone slightly increases the occupancy of the intermediate FRET states, but it does not increase the occupancy of the highest FRET state that corresponds to the Acc conformation (Fig. 4c, d). This further confirms that L-Trp alone cannot stabilize the active conformation sufficiently to result in the receptor activation,

as probed at the CRD. Importantly, the PAM effect of L-Trp is due to increasing the occupancy of intermediate transition states.

To explore the molecular reasons for the unique outcome of amino acid binding in CaSR compared to mGluRs, we mapped the interaction area of the dimer interface of CaSR between active and inactive conformations (Fig. 5a). A continuous strip of interactions along the LB1 interface of CaSR is maintained in both the active and inactive conformations (Fig. 5a, left). This is in contrast with mGluR1, evolutionary the closest mGluR to CaSR, which shows a smaller and less distributed interaction area in LB1 (Fig. 5a, right). We hypothesized that this difference in the distribution of intermolecular interactions at LB1 could be partly responsible for the distinct outcome of amino acid binding in CaSR compared to mGluRs. To investigate the structural

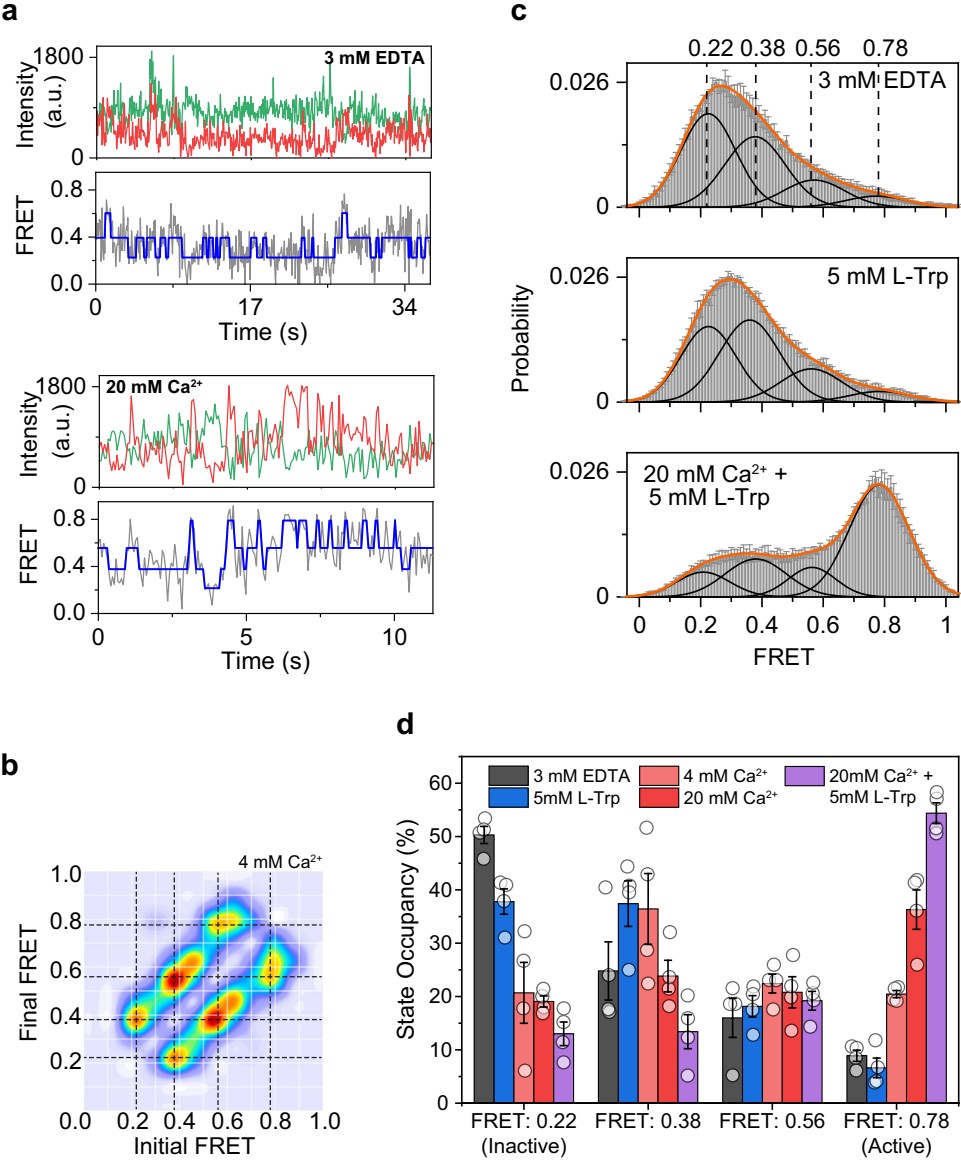

**Fig. 4 CRD of CaSR is in equilibrium between four conformational states. a** Sample single molecule traces of E593UAA in 3 mM EDTA or 20 mM Ca$^{2+}$ showing donor (green) and acceptor (red) intensities, corresponding FRET (gray), and idealized FRET trajectory from HMM fit (blue). **b** Transition density plot of E593UAA. Dashed lines represent the most frequently observed transitions and were used for multiple-peak fitting of FRET histograms. **c** smFRET population histogram in the presence of 3 mM EDTA, 5 mM L-Trp, or 20 mM Ca$^{2+}$ and 5 mM L-Trp. Data represent mean ± s.e.m. of $n = 4$ independent biological replicates. Histograms were fit with four single gaussian distributions (black) centered at 0.22, 0.38, 0.56, and 0.78, and the cumulative fit is overlaid (orange). **d** Occupancy of the four FRET states of the CRD in the presence of increasing ligand concentrations. Values represent mean ± s.e.m. area under individual FRET peaks from $n = 4$ independent biological replicates.

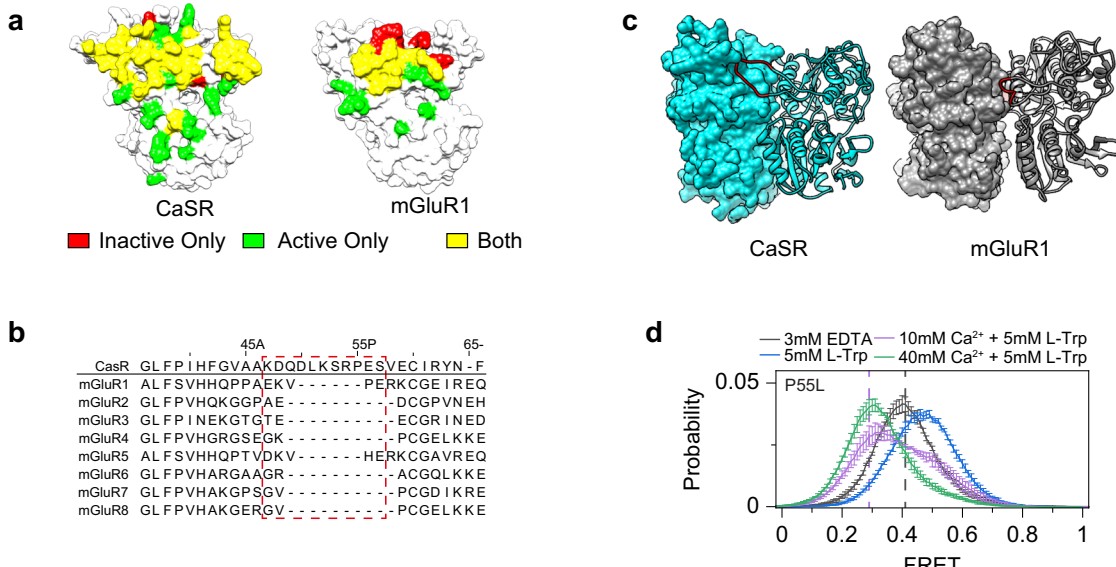

**Fig. 5 An elongated interprotomer is critical for VFT coordination. a** Surface representation of CaSR (left, PDB 5K5S) and mGluR1 (right, PDB 1ISR) showing contacts made between monomers in the inactive structure only (red), active structure only (green), or both (yellow). **b** Multiple sequence alignment of human CaSR and mGluRs. CaSR is used as reference for residue numbering. Sequence of interprotomer loop indicated by red dashed line. **c** Surface and ribbon representation of CaSR (left, PDB 5K5S), and mGluR1 (right, PDB 1ISR) with interprotomer loop colored red. **d** smFRET population histogram for P55L in the presence of 3 mM EDTA, 5 mM L-Trp, 10 mM $Ca^{2+}$ and 5 mM L-Trp, or 40 mM $Ca^{2+}$ and 5 mM L-Trp. For reference, dashed lines indicate centers of wild-type distributions for EDTA (gray) and 10 mM $Ca^{2+}$ + 5 mM L-Trp (purple). Data represent mean ± s.e.m. of $n = 3$ independent biological replicates.

elements underlying this, we compared the sequence of LB1 in CaSR and mGluRs (Fig. 5b). We detected a conserved structured loop which is significantly elongated in CaSR compared to mGluRs (Fig. 5c). In CaSR, this loop was observed in crystal structures and was suggested to be involved in receptor dimerization[13]. This loop extends from one protomer and makes extensive contacts with and docks into the adjacent promoter through a highly conserved sequence (Supplementary Fig. 4a). Specifically, the highly conserved residue P55 makes contacts with Y161 and W458 in the adjacent promoter (Supplementary Fig. 4b). Multiple disease-associated mutations localize to this interprotomer loop including the P55L mutation[49,50]. We hypothesized that this loop, by restricting the movement of protomers through increased LB1 dimer interactions, could be part of the reason for smaller conformational rearrangement of CaSR compared to mGluR2 (Fig. 1). Interestingly, the loop is disordered in a recent structure of a nanobody stabilized inactive CaSR that positioned the VFT domain in an mGluR-like inactive conformation[15]. To test this, we performed smFRET experiment on a D451UAA construct with P55L mutation in the presence of different ligand conditions. First, in the absence of ligands, we observed the same 0.41 inactive peak as wildtype CaSR (Fig. 5d). Next, we detected a similar active peak as wildtype (Fig. 5d) but at a much higher calcium concentration (40 mM $Ca^{2+}$ + 5 mM L-Trp) consistent with the physiological loss-of-function phenotype for his mutant. Surprisingly, in the presence of 5 mM L-Trp alone the FRET distribution shifted to a new stable state centered at FRET = 0.5 (Fig. 5d, blue). The occupancy of this new FRET state decreased by adding increasing concentrations of $Ca^{2+}$. This suggests the loop controls the conformational space of L-Trp bound CaSR. Functional characterization showed that L-Trp still functions as a PAM for the P55L mutant (Supplementary Fig. 4c). To further investigate the role of this loop in amino acid modulation of $Ca^{2+}$ potency, we truncated the loop to be more mGluR-like (Δ47–57) and quantified the effect of 10 mM L-Trp on $Ca^{2+}$ potency for the new mutant. We observed a 40%, 45%,

and 58% reduction of $Ca^{2+}$ $EC_{50}$ for wild-type, P55L, and Δ47–57, respectively (Supplementary Fig. 4c). The larger reduction of $EC_{50}$ is indicative of amino acids becoming a more critical ligand for receptor activation when interprotomer loop contacts are disrupted. Together, these results are consistent with the interpretation that this extended loop shapes the energy landscape of CaSR by restricting the movement of the LB1 and reduce the overall contribution of amino acid binding to receptor conformational rearrangement and activation.

**Negative charge density of the dimer interface is a key regulator of CaSR activation.** To further investigate the structural determinants of the CaSR and mGluR adaptation for their physiological ligands we performed a multiple sequence alignment of CaSR, the mGluRs, and the taste receptors (Tas1Rs) (Supplementary Fig. 5a). We observed a significantly higher number of negatively charged residues on the lower lobe interface (LB2) in CaSR compared to other class C GPCRs (Supplementary Fig. 5a). This was previously highlighted in crystal structures and was suggested to be important for ion binding[13]. We quantified the electrostatic potential of CaSR and all other class C GPCRs with published structures (Fig. 6a and Supplementary Fig. 5b) and found that the LB2 of the CaSR VFT domain is significantly more electronegative compared to all other class C GPCRs[14,36] (Fig. 6a and Supplementary Fig. 5b). To identify the functionally significant amino acids within this interface, we aligned CaSR and 200 orthologs (Fig. 6b). This analysis revealed that the negative surface charges on the LB2 of CaSR are very conserved across different organisms (Fig. 6b) emphasizing the universal functional role of this design. Notably, these surface charges occur in three distinct patches within a structurally conserved helix–sheet–helix motif that, in human, includes a DDD motif (I), a EKFREEAERD motif (II), and a DEEE motif (III) (Fig. 6b). Importantly, this topology is absent in other class C GPCRs (Supplementary Fig. 5a). Furthermore, many disease-associated mutations in CaSR such as D215G[51], R220W[52], R227L[49], R227Q[51], and

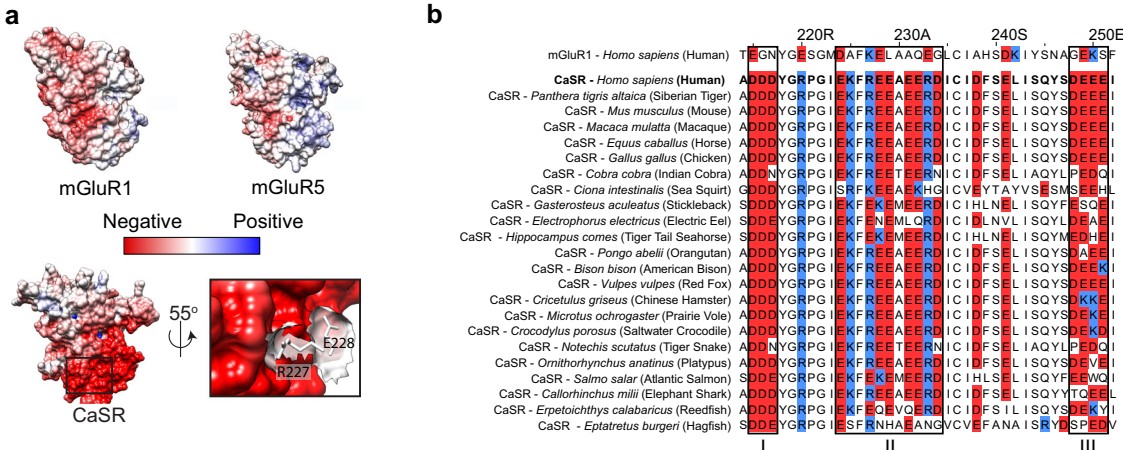

**Fig. 6 The highly negative charge of the CaSR dimer interface is variable. a** Electrostatic potential map of mGluR1 (top left, PDB 1ISR), mGluR5 (top right, PDB 3LMK), and CaSR (bottom left, PDB 5K5S). Close up inset of CaSR (bottom right) showing stick and surface representation of R227 and E228 at the lower lobe interface. **b** Multiple sequence alignment of mGluR1, CaSR, and select CaSR orthologs with negatively charged residues (red) and positively charged residues (blue) highlighted. CaSR is used as reference for residue numbering. Boxes indicate regions of high charge density unique to CaSR.

E228K[53] localize to these structural elements highlighting their importance in CaSR function. Because the LB2 surface of the monomers come into proximity in the active state (Fig. 5a), we hypothesized that the negative charge density of the intermolecular interface of LB2 modulates the stability of the active state via electrostatic repulsion. To test this, we performed smFRET experiments on mutants that alter this surface charge density. Specifically, we tested R227L and E228K mutants within the EKFREEAERD segment which increase and decrease the surface negative charge, respectively. Both are disease-associated mutations in humans that result in loss-of-function (for R227L) or gain-of-function (for E228K) in CaSR and cause hyperparathyroidism and hypocalcemia, respectively[49,53]. In smFRET experiments, both variants showed the same canonical FRET states as wildtype (Fig. 7a), suggesting that the mutations do not alter the overall coordination of protomers within the dimer. However, we found that in the presence of 10 mM $Ca^{2+}$, E228K had substantially higher occupancy of the active state while R227L had lower occupancy of the active state, compared to the wildtype CaSR (Fig. 7b). This observation is consistent with the gain-of-function and loss-of-function phenotypes for these mutants and our assignment of the active and inactive FRET conformations. Importantly, these residues are not part of known $Ca^{2+}$ binding sites[12–15] (Supplementary Fig. 5d) and therefore this observed effect on receptor sensitivity is allosteric. Interestingly, the effect of these mutations on the occupancy of active state relative to wild-type CaSR is the same for L-Trp only or $Ca^{2+}$ only (Fig. 7b), suggesting that binding of both ligands likely converge on the same downstream conformational pathway. The fact that increasing the negative charge of LB2 impedes and decreasing the negative charge facilitates activation of CaSR supports our interpretation that the very high negative charge density of LB2 in CaSR has evolved to control activation via electrostatic repulsion. This arrangement also limits spontaneous activation of CaSR via limiting the occupancy of active state in the ligand free receptor. This also could explain why spontaneous visits to the active state in the ligand free condition cannot result in receptor activation.

To explore the possible evolutionary role of the above design we quantified the degree of variation in the charge distribution on LB2 in CaSR and mGluR1 in different organisms. We selected the location of charged residues in the LB2 interface of CaSR and mGluR1 from 200 organisms and mapped their conservation

onto the canonical structure of human CaSR and mGluR1 (Fig. 7c). Our analysis showed that the LB2 of CaSR is significantly more permissive to the gain and loss of charged residues at each position than mGluR1. In other words, throughout evolution the location of charges is more mobile in CaSR compared to mGluR1 (Fig. 7c and Supplementary Fig. 6a). As a comparison, this level of variability does not exist in LB1 of CaSR or mGluR1 (Supplementary Fig. 6b). We experimentally verified the ability of interface charge to modulate sensitivity with previously uncharacterized mutations E249K, E251K, and V258R, which are in the most variable region of LB2 (Supplementary Fig. 6a) and are not near any implicated ion binding sites from structures (Supplementary Fig. 5d). All mutations reduce the negative surface charge density and showed increased calcium sensitivity compared to wild-type CaSR (Fig. 7d). Finally, we performed smFRET experiments on WT CaSR and in the presence of increasing concentration of NaCl. We found that FRET distribution shifted further to the left in the presence of 5 mM L-Trp only when the concentration of NaCl increased (Fig. 7e and Supplementary Fig. 5e) further suggesting the role of electrostatic repulsion in regulating occupancy of the 0.29 FRET state. The ability of NaCl to modulate the effect of L-Trp and in the absence of $Ca^{2+}$ is consistent with our interpretation that amino acid binding induces a partial or transient engagement of the LB2–LB2 interface. Moreover, it suggests that the regulatory effect of the negatively charged LB2 patch is independent of $Ca^{2+}$. These data and the large variability of charge distribution on LB2 of CaSR among different organisms and the fact that the negative charge on the LB2 is a key controller of receptor activation raises the possibility that this electrostatic interface may have evolved as a mechanism for tuning the receptor sensitivity and the setpoint of CaSR activation, to match the needs of different organisms.

We next asked how disease-associated mutations in CaSR distort the canonical structure and conformational dynamics of the receptor to cause the pathological effect. There are over 200 known disease-associated mutations in CaSR that generally cause hypercalcemia, hyperparathyroidism, hypocalcemic hypercalciuria, and Bartter syndrome type V[29,50,54,55]. Many of these mutations map to the extracellular dimer interface and away from the residues that are known to be involved in ligand coordination and therefore exert their effect allosterically (Fig. 8a). We tested several of these mutations that either sensitize (reduce $EC_{50}$) or desensitize (increase $EC_{50}$) the receptor to $Ca^{2+}$. Characterization

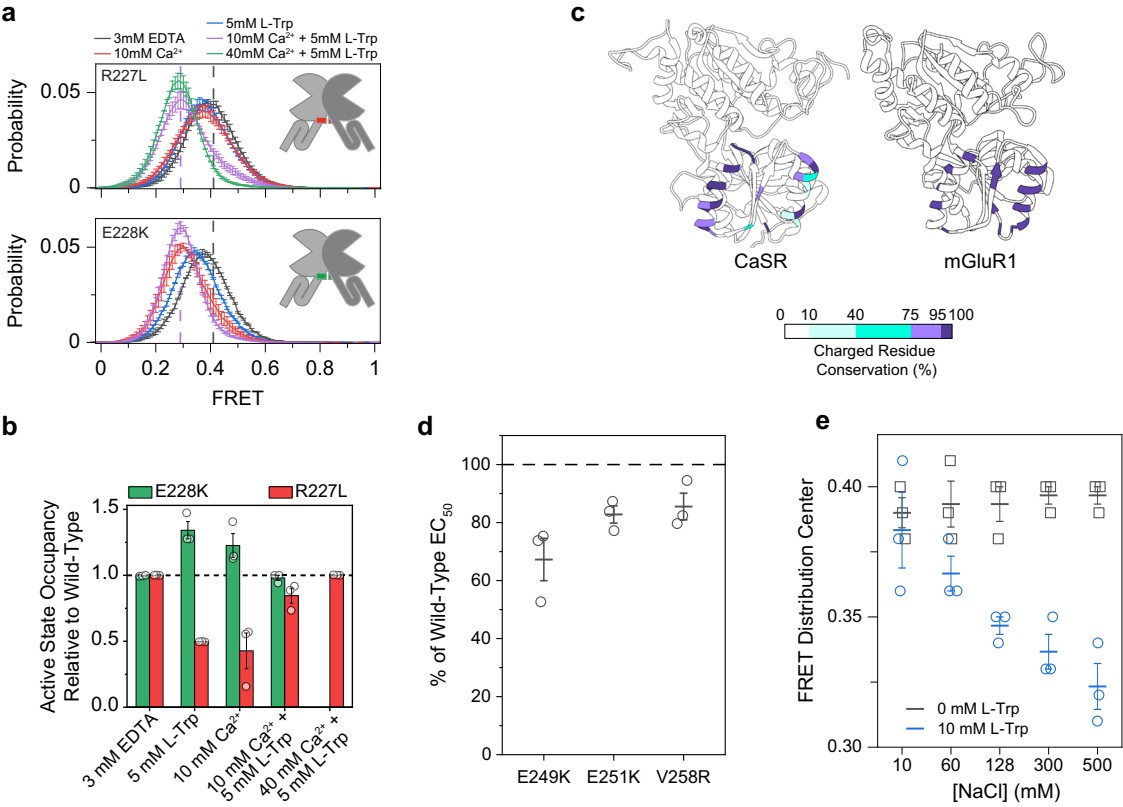

**Fig. 7 Relative charge distribution of an electrostatic interface tunes calcium sensitivity of CaSR. a** smFRET population histograms of R227L and E228K in the presence of 3 mM EDTA, 5 mM L-Trp, 10 mM $Ca^{2+}$, 10 mM $Ca^{2+}$ and 5 mM L-Trp, or 40 mM $Ca^{2+}$ and 5 mM L-Trp. For reference, dashed lines indicate centers of wild-type distributions for EDTA (gray) and 10 mM $Ca^{2+}$ + 5 mM L-Trp (purple). Data represent mean ± s.e.m. of $n = 3$ independent biological replicates. **b** Occupancy of the active FRET state of R227L and E228K for each condition normalized to wild-type. Values represent mean ± s.e.m. area under active FRET peaks from smFRET population histograms, averaged over three independent biological replicates, centered at 0.41 (inactive) and 0.29 (active). 40 mM $Ca^{2+}$ + 5 mM L-Trp condition was not tested for E228K **c** Ribbon representation of CaSR (left) and mGluR1 (right) (PDB IDs: 5K5S and 1ISR) displaying the conservation of charged residues at each position in LB2 across 200 species. **d** $EC_{50}$ for E249K, E251K, V258R CaSR as a percentage of wild-type. Data represents the mean ± s.e.m. of $n = 3$ independent biological replicates. **e** Center of a single gaussian distribution fit to FRET histograms. Data represents the mean ± s.e.m. of $n = 3$ fits to independent biological replicates.

of SNAP-tagged mutants by smFRET showed that, while all the tested mutants can signal (Supplementary Fig. 7a, b) with altered $EC_{50}$, they adopt a diverse range of architecture at the extracellular domain in their active and inactive states, as measured from smFRET histograms (Fig. 8b and Supplementary Fig. 7c). Moreover, we found no correlation between the physiological effect of a mutation and how FRET distribution changed compared to the canonical receptor (Fig. 8b). For example, C129S[56] and C131G are both sensitizing mutations, but they have opposite effect on the FRET distribution. However, we found that all sensitizing mutations that we tested showed a reduction in the cross-correlation amplitude in the absence of ligand compared to the wildtype CaSR (Fig. 8c). By contrast, the cross-correlation amplitude of desensitizing mutations was higher in the presence of $Ca^{2+}$ (Fig. 8c). Overall, these results reveal a large degree of permissiveness in CaSR dimer architecture that was previously unknown and indicate a complex relationship between subtle sequence variations, coordination of the ECD of CaSR, and the receptor sensitivity.

## Discussion

Receptor dimerization is a powerful design principle for initiating signal transduction. In this paradigm, ligand binding to monomeric receptors triggers dimerization or oligomerization of receptors, which then results in activation of intracellular enzymatic or signaling domains. However, while this model has been shown to be true to the first approximation, it does not explain how the signaling output of the receptor can be tuned by native or artificial ligands[57,58]. In this research we discovered a general mechanism through which the activation setpoint of CaSR, a dimeric class C GPCR, is finely tuned. We propose that this design is potentially prevalent in other dimeric receptors.

Our findings revealed the design of CaSR activation, which has evolved to measure concentration of extracellular $Ca^{2+}$. Despite very high overall structural similarities between CaSR and mGluRs (1.7 Å RMSD), these receptors evolved to sense signals with very different temporal profiles. In the case of mGluRs, synaptic glutamate is a signal that varies stepwise between very low to very high levels and over a fast timescale while CaSR must measure slow changes in extracellular $Ca^{2+}$ that varies around ~2.4 mM over hours timescale. Signals preexist their receptor. Therefore, evolution of new receptors requires modification of the ancestral receptor to not only sense the chemical identity of the new signal, but also to adapt to its temporal structure in native context. Thus, a signal that changes over milliseconds timescale imposes different evolutionary mechanical constraints on the receptor than a signal that changes over hours timescale.

We found that several unique structural features of CaSR, compared to mGluRs, can partially account for the distinct activation mechanism of CaSR. First, an elongated intermolecular loop previously visualized in crystal structures and involved in linking the LB1s of CaSR[13] and second, an increased contact area

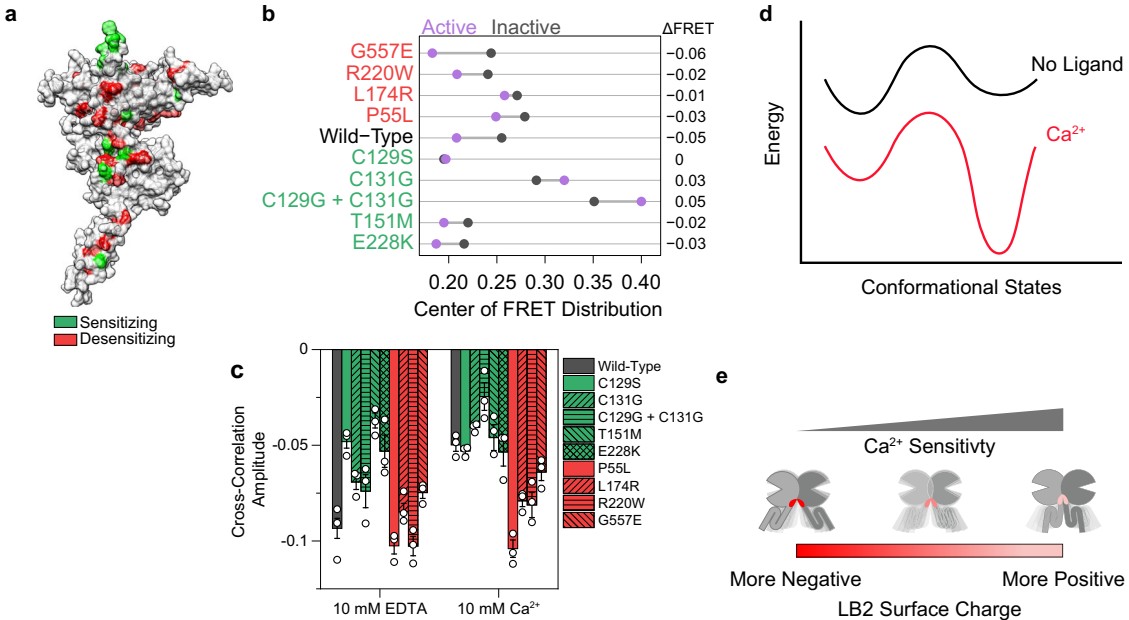

**Fig. 8 Changes in CaSR dynamics by mutation predict calcium sensitivity. a** Surface representation of CaSR (PDB 5K5T) with known sensitizing (green) and desensitizing (red) mutations mapped to the structure. **b** Barbell plot showing the center of the FRET distribution in the inactive condition (10 mM EDTA, gray) and active condition (10 mM $Ca^{2+}$ + 5 mM L-Trp, purple) for CaSR mutants. Mutants that desensitize (red) or sensitize the receptor (green) are grouped together. The amplitude of the FRET shift is shown on the right. **c** Amplitude of receptor cross correlation of wild-type (gray), sensitizing (green) mutations, and desensitizing (red) mutations in the presence of 10 mM EDTA or 10 mM $Ca^{2+}$. Data represent mean ± s.e.m. of $n = 3$ independent biological replicates. **d** schematic conformational energy landscape of CaSR in the absence of ligand (black) and in the presence of $Ca^{2+}$ (red). **e** Model of CaSR illustrating how the LB2 surface charge modulates sensitivity. As the LB2 becomes more positive, the stability of the active interface increases resulting in increased receptor sensitivity.

for the LB1 interface. These features likely restrict movement of the ECD in CaSR and holds the LBDs closer, resulting in a unique activation mechanism. Our results suggest that the energy barrier between the active and inactive states is lower in CaSR than in mGluR2 (Fig. 8d) which enables CaSR to visit and sample the active state often and briefly, even in the absence of any ligand. However, the large focused negative electrostatic surface charge of LB2 in CaSR impedes activation via electrostatic repulsion of the LB2 domains, which must come in proximity for receptor activation (Fig. 8e). This likely prevents spontaneous activation of CaSR in the absence of ligand or by L-amino acids alone. Binding of amino acids has been shown in structure models to induce closure of the VFT domain but not engagement of the LB2 interface that resemble the active conformation. Our results suggest that amino acids also induce conformational changes in the VFT that may include temporary or partial engagement of the LB2 interface. We observed the ability of L-Trp to induce an increase in occupancy of the 0.29 FRET active state for D451UAA, but we did not observe a similar increase in the occupancy of the 0.78 FRET active state for E593UAA which probes the CRD. This is possibly due to the loose coupling between these domains, as was also observed in mGluR2[40]. Because of this, we are not able to conclude that amino acid binding alone induces the Acc conformation characterized by engagement of both the LB2–LB2 interface and the CRD–CRD interface (Supplementary Fig 8a). Finally, it is possible additional intermediate states exist that we did not resolve due to the spatial and temporal limitations of smFRET measurements.

Existence of the negatively charged patch as an electrostatic control hub for activation could explain how CaSR can function as a salinity sensor as proposed in euryhaline fishes[26,59]. Interestingly, we showed that, between different species, the charge distribution on the LB2 of CaSR is extremely variable, much more

than mGluRs or the LB1 of CaSR. We showed how this design element of CaSR enables tuning the sensitivity of the receptor. Evolution could have used this design feature to optimize CaSR to suit the environmental niche of an animal through discrete evolutionary changes in the charge density of the dimeric interface. Variations of ligand binding pocket to tune ligand affinity is limited and does not map a large phase space. However, the electrostatic tuning mechanism that we described is generalizable and maps the phase space of interface charge distribution to the phase space of receptor sensitivity. Finally, while none of the atomic structures show $Ca^{2+}$ binding at the negative patch on LB2 (Supplementary Fig. 5d), it is possible that transient or weak binding of $Ca^{2+}$ at this region could contribute to CaSR activation via screening the negative charges. This interpretation is consistent with previous reports based on mutagenesis and fragment analysis[13,60,61] and positively charged residues could emulate this proposed effect.

We found that many disease-associated mutations considerably distort the arrangement of the ECD of CaSR without abolishing its ability to signal (with altered $EC_{50}$). This result indicates that the ECD conformation of CaSR is promiscuous and tolerates a range of alternative topologies. This conformational flexibility was likely used throughout evolution of class C GPCRs to explore the structural landscape. Furthermore, modifying the topology of the extracellular domain of CaSR to compensate for the effect of disease-associated mutations may be utilized for drug-based therapies. Such therapeutic strategy could potentially be applied to similar bilobed shaped receptors such as other class C GPCRs and the ionotropic glutamate receptors.

## Methods
**Molecular cloning**. A mouse CaSR construct with a C-terminal FLAG-tag in pcDNA3.1⁺ expression vector was purchased from GenScript (ORF clone:

OMu14241D) and validated by sequencing (ACGT). A SNAP-tag (pSNAP$_f$, NEB) flanked by GGS linkers was inserted at position 21 using HiFi DNA Assembly Master Mix (NEB). Point mutations in CaSR (P55L, C129S, C131G, C129G + C131G, T151M, L174R, R220W, R227L, E228K, G557E) were introduced using a QuikChange site-directed mutagenesis kit (Qiagen). The SNAP-tag CaSR construct was used as the template for mutation of amino acid E593 to an amber codon (TAG) via QuikChange mutagenesis. AscI restriction sites were used to insert mEGFP (mEGFP-N1, gift from Michael Davidson (Addgene plasmid # 54767) at the C-terminus of the GenScript construct, and it was used as the template for mutation of amino acid D451 to an amber codon (TAG) via QuikChange mutagenesis. All plasmids were verified by sequencing (ACGT). DNA restriction enzymes, DNA polymerase, and DNA ligase were purchased from New England Biolabs. Plasmid preparation kits were obtained from Macherey-Nagel.

**Cell culture conditions**. Cells (HEK293T) purchased from Sigma were maintained in culture media consisting of: high glucose DMEM (Corning), 10% (v/v) fetal bovine serum (GE Healthcare), 100 unit/mL penicillin–streptomycin (Gibco), and 10 mM HEPES (pH 7.4, Gibco). Cells were incubated at 37 °C under 5% $CO_2$ during maintenance. A 0.05% trypsin–EDTA solution (Gibco) was used to passage cells. For unnatural amino acid-containing protein expression, the growth medium was supplemented with 0.65 mM 4-azido-L-phenylalanine (Chem-Impex International). All media were filtered by 0.2 µM aPES filters (Fisher Scientific).

**Transfection and protein expression**. At 24 h before transfection, HEK293T cells were cultured on 18-mm polylysine-coated glass coverslips (VWR). For unnatural amino acid labeling, one hour before transfection, cell culture medium was changed to growth medium supplemented with 0.65 mM 4-azido-L-phenylalanine. CaSR plasmids with amber codons as described above and pIRE4-Azi plasmid (pIRE4-Azi was a gift from I. Coin, Addgene plasmid no. 105829) were co-transfected (1:1 w/w) into cells using Lipofectamine 3000 reagents (Invitrogen) (total plasmid: 1.6 µg per 18-mm coverslip). Growth medium containing 0.65 mM 4-azido-L-phenylalanine was refreshed after 24 h and the cells were grown for another 24 h (total 48 h expression). On the day of the experiment, 30 min before labeling, 4-azido-L-phenylalanine supplemented growth medium was removed, and cells were washed twice by extracellular buffer solution containing 128 mM NaCl, 2 mM KCl, 2.5 mM CaCl$_2$, 1.2 mM MgCl$_2$, 10 mM sucrose, 10 mM HEPES, and pH 7.4 and were kept in growth medium without 4-azido-L-phenylalanine. Before the addition of labeling solution (below), cells were washed once with extracellular buffer solution. For SNAP-tag experiments, CaSR plasmids as described above were transfected into cells using Lipofectamine 3000 (Invitrogen) (1 µg per 18-mm coverslip).

**SNAP-tag labeling in live cells**. For SNAP labeling of the N terminus VFT domain, cells were incubated with 4 µM of benzylguanine Alexa-647 (NEB) and 4 µM of benzylguanine DY-549P1 (NEB) in extracellular buffer for 30 min at 37 °C. To remove excess dye after labeling, the coverslip was gently washed twice in extracellular buffer.

**Unnatural amino acid labeling in live cells by azide-alkyne click chemistry**. A modified version of previously reported protocols[45,46] was used to label the incorporated 4-azido-L-phenylalanine in live cells. For labeling, the following stock solutions were made: Cy3 and Cy5 alkyne dyes (Click Chemistry Tools) 10 mM in DMSO, BTTES (Click Chemistry Tools) 50 mM, copper sulfate (Millipore Sigma) 20 mM, aminoguanidine (Cayman Chemical) 100 mM and (+)-sodium L-ascorbate (Millipore Sigma) 100 mM in ultrapure distilled water (Invitrogen). In 656 µl of extracellular buffer solution, Cy3 and Cy5 alkyne dyes were mixed to a final concentration of 20 µM and 16 µM of each, respectively. A solution of fresh copper sulfate solution and BTTES (1:5 molar ratio) was premixed and then added to a final concentration of 150 and 750 µM, respectively. Next, aminoguanidine was added for a final concentration of 1.25 mM and (+)-sodium L-ascorbate was added for a final concentration of 2.5 mM. The total labeling volume was 0.7 mL. The completed labeling mixture was kept at 4 °C for 8 min followed by 2 min at room temperature and kept in darkness before addition to the cells. Cells were washed before addition of the labeling mixture. During labeling, cells were kept in the dark at 37 °C and 5% $CO_2$ for 10–15 min. Post labeling, extracellular buffer was used to wash the cells twice.

**Single-molecule FRET measurements**. Single-molecule FRET experiments were performed using flow cells prepared with glass coverslips (VWR) and slides (ThermoFisher Scientific) passivated with mPEG (Laysan Bio) and 1% (w/w) biotin-PEG to prevent nonspecific protein adsorption as previously described[39,62]. Before experiments, flow cells were prepared by first incubating with 500 nM NeutrAvidin (ThermoFisher Scientific) for 5 min followed by either 20 µM biotinylated FLAG antibody (A01429, GenScript) or 20 µM biotinylated GFP antibody (ab6658, Abcam) for 30 min. Washing removed unbound NeutrAvidin and biotinylated antibody. Washes and protein dilutions were done with T50 buffer (50 mM NaCl, 10 mM Tris, pH 7.4).

Post labeling, cells were recovered from 1 to 3 18-mm polylysine-coated glass coverslips (VWR) by incubating with DPBS (Ca$^{2+}$-free, Gibco) followed by gentle

pipetting. After resuspension, cells were briefly kept on ice and then pelleted by centrifugation at 4000 × $g$ at 4 °C for 10 min. The cell pellet was lysed in 100–300 µl (depending on the number of coverslips transfected) of a lysis buffer consisting of 10 mM Tris, 150 mM NaCl, protease inhibitor tablet with EDTA (ThermoFisher Scientific) and 1.2% IGEPAL (Sigma), pH 7.4. After 1 h of gentle mixing at 4 °C, lysate was centrifuged at 20,000 × $g$ at 4 °C for 20 min. The supernatant was collected and diluted (two- to tenfold dilution depending on the concentration) and was then added to the flow chamber to achieve sparse surface immobilization of labeled receptors via their C-terminal tag (mEGFP or FLAG). Sample dilution and washes were done using a dilution buffer consisting of 10 mM Tris, 150 mM NaCl, and 0.05% IGEPAL (Sigma), pH 7.4. The flow chamber was washed extensively with dilution buffer to remove unbound proteins (>20× chamber volume) after optimal receptor density was achieved. Finally, labeled receptors were imaged in imaging buffer consisting of 4 mM Trolox, 128 mM NaCl, 2 mM KCl, 40 mM HEPES, 0.05% IGEPAL and an oxygen-scavenging system consisting of 4 mM protocatechuic acid (Sigma) and 1.6 U/mL bacterial protocatechuate 3,4-dioxygenase (rPCO) (Oriental Yeast), pH 7.35. All chemicals were purchased from Sigma or Millipore. All buffers were made using ultrapure distilled water (Invitrogen) or ultrapure filtered water (Milli-Q).

For the extended wash, a dataset was acquired prior to passing 50× chamber volume of wash buffer through the flow chamber every 30 min for 2 h. After the fifth wash at 2 h, a dataset was acquired to compare to the pre-wash data. Finally, a dataset was acquired in the presence of 10 mM Ca$^{2+}$. Samples were imaged with a ×100 objective (Olympus, 1.49 numerical aperture, oil-immersion) on a TIRF microscope in the oblique illumination mode and using an excitation filter set with a quad-edge dichroic mirror (Di03-R405/488/532/635, Semrock) and a long-pass filter (ET542lp, Chroma), with 30 ms time resolution unless stated otherwise. Lasers at 532 and 638 nm (RPMC Lasers) were used for donor and acceptor excitation, respectively.

**smFRET data analysis**. Analysis of single-molecule fluorescence data was performed using smCamera (http://ha.med.jhmi.edu/resources/), custom MATLAB (MathWorks) scripts, custom Python scripts, and OriginPro (OriginLab). The selection of particles and generation of raw FRET traces was done automatically within the smCamera software. Only particles that showed acceptor signal upon donor excitation, acceptor brightness greater than 10% above background, and a Gaussian intensity profile were automatically selected (Supplementary Fig. 8b). Donor and acceptor intensities were measured over all frames for the selected particles. Only particles that showed a single donor and a single acceptor bleaching step during the acquisition time (Supplementary Fig. 8c), stable total intensity ($I_D + I_A$), anticorrelated donor and acceptor intensity behavior without blinking events and that lasted for more than 3 s were manually selected for further analysis (~10%–15% of total molecules per movie). Two individuals independently analyzed subsets of the data and the results were compared and shown to be identical. Furthermore, a subset of data was analyzed blind to ensure no bias in the analysis. Apparent FRET efficiency was calculated as $(I_A - 0.085 \times I_D)/(I_D + I_A)$, where $I_D$ and $I_A$ are raw donor and acceptor intensities, respectively. Every experiment was repeated in triplicate to ensure reproducibility of the results unless otherwise noted. A minimum of 300 FRET traces from three independent biological replicates were used to generate population smFRET histograms unless otherwise stated. Before trace compilation, FRET histograms of individual particles were normalized to 1 to ensure that each trace contributed equally, regardless of trace length. Error bars on histograms represent the standard error three independent biological replicates.

Peak fitting of smFRET histograms was performed using OriginPro with either 1, 2 or 4 Gaussian distributions as

$$y(x) = \sum_{i=1}^{n} \frac{A_i}{w_i \sqrt{\frac{\pi}{2}}} e^{-2\frac{(x-x_{ci})^2}{w_i^2}} \qquad (1)$$

where $n$ is the number of Gaussians, $A$ is the peak area, $xc$ is the FRET peak center and $w$ is the full width half maximum for each peak. Peak centers ($xc$) were constrained as mean FRET efficiency of a conformational state ±0.02. The mean FRET efficiencies associated with different conformational states was determined based on the most frequent transitions between FRET efficiencies in transition density plots, which are denoted by dashed lines (Figs. 2f, 3d, 4b). Peak fitting used the LevenBerg–Marquardt algorithm to determine the best fit by Chi-square with a tolerance of 1E−9 in OriginPro. This analysis is further described below. Peak widths were constrained as $0.1 \leq w \leq 0.24$. Peak areas were constrained as $A > 0.001$. Probability of state occupancy was calculated as area of specified peaks relative to the total area, which is defined as the cumulative area of all individual peaks.

Raw donor, acceptor, and FRET traces were idealized by fitting with a hidden Markov model (HMM) using ebFRET software[63]. Traces for which a single state was assigned were omitted from downstream analysis. Transition density plots were then generated by extracting all the transitions where ΔFRET > 0.1 from the idealized traces.

The cross-correlation (CC) of donor and acceptor intensity traces at time $\tau$ is defined as (2) $CC(\tau) = \delta I_D(t)\delta I_A(t + \tau)/(\langle I_D \rangle + \langle I_A \rangle)$, where (3) $\delta I_D(t) = I_D(t) - \langle I_D \rangle$, and (4) $\delta I_A(t) = I_A(t) - \langle I_A \rangle$. $\langle I_D \rangle$ and $\langle I_A \rangle$ are time average donor and acceptor intensities, respectively. Cross-correlation calculations were performed on the same traces used to generate the histograms. Cross-correlation data were fit

with a single exponential decay function (5) $y(x) = y_0 + A \cdot e^{-\frac{x}{t}}$ by OriginPro (OriginLab).

**Calcium mobilization assay.** Coverslips with HEK293T cells expressing N-terminal SNAP CaSR were briefly washed in extracellular buffer solution before they were placed into 600 µL of extracellular buffer solution with 4 µM Oregon Green Bapta-1 (OGB1) AM (ThermoFisher) and 2 µM of benzylguanine Alexa-647(NEB). During labeling, cells were kept in the dark at 37 °C and 5% $CO_2$ for 30 min. Cells were washed twice after labeling with extracellular buffer solution to remove excess dye and were transferred to a flow chamber (Chamlide) for live-cell confocal imaging. The flow chamber was attached to a gravity flow control system (ALA Scientific Instruments) to switch between buffer application during experiments. Buffers were applied at the rate of 5 mL/min. A zero-calcium buffer (128.75 mM NaCl, 2 mM KCl, 1 mM $MgCl_2$, 0 mM $CaCl_2$, 20 mM HEPES, 5.5 mM D-glucose, pH 7.4) and a high-calcium buffer (68.75 mM NaCl, 2 mM KCl, 1 mM $MgCl_2$, 40 mM $CaCl_2$, 20 mM HEPES, 5.5 mM D-glucose, pH 7.4) were mixed to achieve desired $CaCl_2$ concentrations while maintaining constant osmolarity.

Time-series data was collected using a Zeiss Axio Observer 7 inverted confocal microscope equipped with an LSM800 GaAsP-PMT detectors and a Plan-Apochromat ×40 objective (Zeiss, 1.3 numerical aperture, oil immersion) and the supplied Zen Blue software (2.3 system). A pixel size of 0.312 µm × 0.312 µm resulted in field of view 319.45 µm × 319.45 µm. 488 and 640 nm lasers were used to excite OGB1 and Alexa-647 dyes, respectively. Data was acquired at ~1 Hz.

Movies were analyzed in Fiji[64] by manually drawing a region of interest (ROI) centered on individual cells that showed labeling with Alexa-647 dye (50–100 cells per field of view resulting in a minimum of 150 cells across three individual biological replicates) indicating cells expressing CaSR. Cells without labeling did not respond to changes extracellular calcium. Built-in Fiji functions were used to calculate the integrated intensity for the ROI of the OGB1-AM signal over all frames. Cell response profiles were visualized and normalized by $y(t) = I_t - I_{min} / I_{max} - I_{min}$ where $I_t$ is the intensity at time $t$, and $I_{max}$ and $I_{min}$ are the maximum and minimum values for the cell response profile using Rstudio and custom R scripts. Cells that did not respond to extracellular calcium or showed drift were discarded (10–25% of all cell ROIs). Response profiles of individual cells were summed and treated as a single ROI before quantification of response and fitting of a dose-response curve. Cellular response was quantified by integrating the response curve during application of extracellular calcium for each concentration. Dose–response curves were calculated using OriginPro (OriginLab) by fitting to $y(x) = A1 +$

$\frac{(A2-A1)}{\left(1+10^{((LOG \times 0 - x)*p)}\right)}$ where $A1$ is the bottom asymptote, $A2$ is the top asymptote, $LOG \times 0$ is the center, and $p$ is the hill slope.

**Analysis of multiple sequence alignments.** CaSR and mGluR2 homologs were pulled from Ensembl[65] (version 103) using the REST API (gene IDs: ENSG00000036828, ENSG00000164082). Two sequences shorter than 600 amino acids were discarded. All alignments were created using MUSCLE[66] and default parameters. The conservation of charged residues (Lys, Arg, Glu, Asp) was defined as: $\frac{\#Charged\ Residues\ Observed}{\#Sequences}$.

**Structural analyses.** Distances were measured in Chimera[67]. Based on the spectral overlap of Cy3 alkyne and Cy5 alkyne, a Förster radius ($R_0$) of 54 Å was used to convert raw FRET efficiency $f$ to an approximate distance using $FRET = 1/(1 + (R/R_0)^6)$.

Prior to the calculation of electrostatic potential, hydrogens were added using the AddH tool in chimera and PROPKA[68,69] with pH of 7.4 was used to determine protonation state of residues. Electrostatic potential maps were calculated in Chimera using APBS[70] and PDB2PQR[71] using the PARSE force field.

Identification of interprotomer contacts was performed in Chimera. Inactive and active structures of CaSR and mGluR1 (PDB: 5K5T, 5K5S, 1EWT, 1ISR) were prepared prior to analysis using the Dock Prep tool. Residues involved in interprotomer contacts were identified using the FindContacts tool with a VDW Overlap threshold of −1 and 0 Å for hydrogen bonding pairs.

RMSD between CaSR and mGluR was calculated by superimposing the two structures (PDB: 7DTV, 6N51) using DeepView (SwissPDB Viewer)[72] using an iterative magic fit, which gave an RMSD of 1.7 Å over 2164 backbone atoms for only those parts where the structures overlayed well for the full-length monomer, and an RMSD of 1.8 Å over 3696 backbone atoms for the full-length dimer.

For comparison of $Ca^{2+}$ binding sites, structures with bound ions, 5K5S, 5FBK, 5FBH, 7M3F, 7M3E, 7DTV, 7DTT, 7E6T, 7DD7, 7DD6, 7DD5, and 7M3G, were super-imposed in Chimera using the MatchMaker tool.

**Reporting summary.** Further information on research design is available in the Nature Research Reporting Summary linked to this article.

## Data availability

The materials and data reported in this study are available from the corresponding author upon reasonable request. Sample single molecule image data of D451UAA has been deposited in the Harvard Dataverse repository at https://doi.org/10.7910/DVN/PKR9SD. The plasmids used in this study are available from the corresponding author upon reasonable request. The PDB accession codes for structures used in this paper are: 5K5S, 5K5T, 7DTW, 7DTU, 7DTV, 7DTT, 1ISR, 1EWT, 3LMK, 6N51, 5FBK, 5FBH, 7M3F, 7M3E, 7E6T, 7DD7, 7DD6, 7DD5, and 7M3G. Ensembl gene IDs used to search for homologous proteins are ENSG00000036828 and ENSG00000164082. Source data are provided with this paper.

## Code availability

The custom codes for single-molecule data analysis are available at https://github.com/vafabakhsh-lab/smfret-analysis. Custom R scripts used for calcium mobilization analysis are available at https://github.com/FafferMcgee/calcium-analysis. Python scripts used to fetch data from the Ensembl database are available from the corresponding author upon reasonable request.

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

## Acknowledgements

We thank B.W. Liauw and H.S. Afsari for technical assistance. This work was supported by the National Institutes of Health grant R01GM140272 (to R.V.) and by The Searle Leadership Fund for the Life Sciences at Northwestern University and by the Chicago Biomedical Consortium with support from the Searle Funds at The Chicago Community Trust (to R.V.). M.R.S. was supported by Training Grant T32GM-008382 from the National Institute of General Medical Sciences (NIGMS).

## Author contributions

M.R.S. and R.V. conceptualized the study. M.R.S. performed plasmid construction, smFRET experiments, calcium mobilization experiments, and data analysis. R.V. assisted with smFRET data analysis. The paper was written by M.R.S. and R.V.

## Competing interests

The authors declare no competing interests.
