## [Peer Review File · Nature Communications]

Mechanism of Sensitivity Modulation in the Calcium-Sensing Receptor via Electrostatic TuningREVIEWER COMMENTS

Reviewer #1 (Remarks to the Author):

Schamber and Vafabakhsh described the conformational dynamics during CaSR activation using single-molecule FRET experiments. The authors utilize single molecule FRET measurement to monitor the dynamic conformation changes of individual CaSR. They compared two approaches for labeling the CaSR with fluorescent donor and acceptor and conclude that it is more sensitive by inserting unnatural amino acid (UUA) at residue 451 in the CaSR structure and label fluorophore at that location. Then they used this type of smFRET sensor to study the VFT arrangement of CaSR and find out the how negative charge density at the interface of CaSR dimer structure affect its sensitivity to agonist thus the regulation of CaSR activation. Different calcium and amino acid ligand binding conditions in the wild type and several diseases related CaSR mutants were studied. They found that the ligand-free receptor was very dynamic and sampled both active and inactive conformations. Agonist binding stabilized the receptor in the active conformation. Amino acid binding shifted the conformation but did not activate the receptor. By contrast, calcium binding alone significantly increased the occupancy of the active state even though the amino acid was required to fully activate the receptor. The P55L mutation in the LB1 loop caused an increase of the amount of calcium ions required to activate the receptor, consistent with the loss of function of the mutation. High density of negatively charged residues on the LB2 dimer interface was identified and predicted to be a key regulator of CaSR activation. Mutations on the interface either impeded or facilitated the activation depending on whether the mutation strengthened or weakened the negative electrostatic potential. The authors also examined the effect of several disease mutations on the activation process.

Although the reported results look interesting and may bring some new insights into CaSR activation, the paper is not ready to publish on Nature Communications at its current stage due to several major issues:

- 1) A major claim of the paper is the discovery of the negatively charged LB2 dimer interface modulating the receptor sensitivity. This interface and its role in CaSR activation through electrostatic repulsion have been discussed previously (for example, Zhang et al., 2016, Science Advances 2:e1600241) and so the finding is not completely new. This manuscript also largely overclaimed “ We found that several features of CaSR have evolved uniquely compared to mGluRs to satisfy the spatial and temporal constraints of extracellular Ca²⁺. First, an elongated intermolecular loop linking the LB1s of CaSR and an increased contact area for the LB1 interface likely restrict movement of the ECD in CaSR and holds the LBDs closer to the active-like conformation, resulting in a unique activation mechanism. missed earlier work about probing calcium binding sites”(line 301- 305). This results have also been discovered previously by Zhang et al., (2016, Science Advances 2:e1600241)
- 2) The authors largely ignored literatures of previously reported calcium binding sites by Yun Huang et al. (JBC, 2005, Biochem 2007) and Zhang et al (2016, Science Advances 2:e1600241). These included a Ca²⁺ binding site with negatively charged residues located at the dimer interface. The section describing “Negative charge density of the dimer interface is a key regulator of CaSR activation”(line 207-247) is largely inaccurate due to lack of understanding of the reported role of calcium binding sites.

3) The full-length structures of CaSR by Cryo-EM (Ling et al., Cell Research 2021; Gao et al., Nature 2021) have established several signature conformational changes defining the CaSR activation, including the opening/closing motion in the VFT cleft and the LB2 interface, as well as the movement of the LB1 loop. However, none of these locations

was chosen to measure the dynamics. The authors need to rationalize their choices of the site to attach the fluorescent probes and confirm that the measured distance changes correlate with the activating conformational changes defined by the Cryo-EM structures.

4) What new here is probably the sequence comparison across homologs. However, this analysis seems to be too simple to establish the evolutionary role of the interface. A more quantitative evaluation on the electrostatic potentials of CaSR homologs and their relationship to the receptor sensitivity is required to validate the authors' prediction.

5) While the smFRET imaging experiments are quite interesting and useful, the authors need to address and clarify following questions:

a. Do you have further evidence support your hypothesis that the difference in the distribution of intermolecular interaction at LB1 is responsible for effect of amino acid binding in CaSR and mGluRs? (For example, with mutations which increase the intermolecular interaction at LB1, mGluRs response to ligand binding in a way more similar to CaSR, or vice versa.)

b. Do you have explanation about: during activation, how can the ligand binding overcome the electrostatic repulsion of the LB2 domains? Especially why binding L-Trp can bring the state from inactive towards active (not fully active).

c. Is the 30ms temporal resolution sufficient to revolve the dynamic transitions between different conformations of CaSR?

d. The authors should provide single molecule images and movies of both donor and acceptor signal in their experiments.

e. How to prove your FRET is single molecule? Usually, this can be simply done by photobleaching experiments. The authors added an oxygen scavenger system to minimize the photobleaching effects. However, by imaging for sufficient long time, e.g. tens of minutes, one should be able to eventually photobleach the single molecule fluorescence. The author needs to provide such information in their Supporting information.

f. Usually, one can acquire very large data set for single molecule imaging. For most of the experiment data in this paper, 3 individual experiments were carried out. It is not clear what exactly does the '3 individual experiments' stands for? Does it mean 3 individual single molecule measurements? Or does it mean 3 trials with many single molecule measurements each time? If only 3 single molecule measurements are accomplished, the authors definitely needs to acquire more data. If it is 3 individual trials, the authors need to make it clear in their statement.

Minor concerns:

1. Missing error bars in Fig. 3d, 4c, 6d. How many experiments did? Need to put error bar. Can the authors explain more about Fig.6d? Why there is no bar for Occupancy of E228K at 40mM Calcium + 5mM L-Trp?
2. For studying the CRD of CaSR, four conformation states are used to describe the FRET distribution (Fig. 4). How do the authors determine the peak positions in their fitting here?
3. For Supplementary Fig. 1a, what cell line do you use? Does this cell line express endogenous CaSR? Do you have results of negative (cell with empty vector transfection) and positive (cells transfected with nonmodified WT CaSR) control? When you say cell imaging, did you monitor single cells or a cell population? How many cells does this N=3 individual experiment include?

Reviewer #2 (Remarks to the Author):

In the manuscript 'Mechanism of Sensitivity Modulation in a Class C GPCR via Electrostatic Tuning', Schamber et al. used single-molecule FRET, sequence analysis and signaling assay to study conformational changes in the calcium-sensing receptor. Specifically, the authors mapped the ligand-independent and ligand-dependent conformational dynamics of the CaSR N-terminal domain. In addition, they identified a negatively charged patch at the dimer interface of CaSR that is involved in fine-tuning the receptor sensitivity toward extracellular Ca²⁺. This study provides new information on how conformational changes propagate along the receptor. Their identification of the structural hub that allosterically controls receptor activity through electrostatic repulsion is an advance in the field.

Here are some issues that need to be addressed before publication.

Major:

1. The authors concluded that amino acids fit the role of allosteric modulators because L-Trp increases transitions between the active and inactive states, but these transitions are brief and do not increase the active state occupancy sufficiently to result in signaling output. On the other hand, 10mM Ca²⁺ alone was able to significantly increase the occupancy of the active state. Is it possible that some amino acids or amino acid analogs are already bound to the receptor and are cooperating with the added Ca²⁺ to improve the occupancy of the active state?

Previous structural studies of the CaSR (Ling et al., 2021; Chen et al., 2021; Gao et al., 2021; Geng et al., 2016; Zhang et al., 2016) all indicated the presence of an amino acid-like compound in Ca²⁺ or Mg²⁺-bound CaS receptor structures. This compound has been identified to be a Trp derivative (Zhang et al., 2016), and it has sufficient affinity for CaSR that it remains bound to the receptor throughout the protein purification steps.

Fig. 3d indicates that L-Trp alone is able to induce or increase the presence of active conformation. The authors also showed that only the combination of 10mM Ca²⁺ and 5mM L-Trp fully shifts the occupancy

of the active state. These data seem to argue that the amino acids are partial agonists, or co-agonists with Ca^{2+} , not allosteric modulators. This would also be in agreement with findings from the earlier structural studies that amino acids bind to the agonist site in the VFT (Ling et al., 2021; Chen et al., 2021; Gao et al., 2021; Geng et al., 2016; Zhang et al., 2016).

2. The authors stated amino acids facilitate VFT rearrangement. Are the authors referring to VFT closure? Does the transient FRET transition induced by L-Trp correspond to a transient LB1-LB2 closure? Are the authors implying that the VFT remains open majority of the time even when L-Trp is bound?

3. Fig. 6e shows that the charge distribution on the surface of LB2 varies across species. The authors stated that this variation is correlated with calcium sensitivity of CaSR. Are the authors implying that low negative density on LB2 results in higher sensitivity to Ca^{2+} (EC_{50}) in some organisms and vice versa? Are there any functional data that support this kind of hypothesis?

Minor:

1. Line 33. Not all the ECDs of Class C GPCRs are covalently linked. An example is the heterodimeric GABAB receptor.
2. Fig. 3a. It is difficult to distinguish the curves for 5mM Trp and 10mM Trp. Are they overlapping?
3. Is Fig. 3b. Both panels are labeled 5mM L-Trp. Is this correct?
4. Please indicate the construct used to generate Fig. 5d both in the figure and in the legend.

Reviewer #3 (Remarks to the Author):

This study maps the millisecond-scale conformational states and dynamics of the extracellular domain of the calcium sensor CaSR, a dimeric class C GPCR, via TIRF smFRET. The single-molecule data is corroborated with signalling assays, protein sequence and mutation analysis to infer mechanistic details about inter-protomer interactions, in particular the role of electrostatic repulsion/attraction in tuning the activation of CaSR receptor. Overall this is a well designed and carried out study and the model proposed by the authors is supported by the experimental evidence. However, I have some questions/concerns about the data analysis and the interpretation:

1. Fig. 1 data and description on page 4; A FRET shift of 0.04 under activating conditions is attributed exclusively to a distance change of 2.5 Å, but no indication is given about error margins as well as other factors, such as the orientational factor in FRET. Furthermore, the cross-correlation analysis (1d) lacks

numbers/lifetimes to support statement like "the receptor is very dynamic in the absence of ligand" and that this was reduced in the presence of agonists. Overall, tables with fitting parameters/errors would serve the authors well to drive their points across. About x-corr analysis, controls for "slow" vs. "fast" state exchange would be very helpful. Similarly, static/dynamic controls for FRET width analysis (1e) would also be beneficial, especially when drawing conclusions about the 5-ms data presented in Fig. S1.

2. Fig. 2 data and description; the D451UAA sensor is indeed more sensitive to FRET changes (0.12 vs. 0.04), which raises the issue why data on the previous construct (Fig. 1) should still be included in the paper. It is not clear how the dynamics inferred from this construct (x-corr lifetimes) compare to previous construct (Fig. 1 and S1), especially at 5-ms resolution. In addition, it is not clear whether/how panel f density plots were (or can be) used to infer the populations of different conformational states of CaSR in different ligand conditions.

3. Fig. 3 data (L-Trp titration); the examples shown in 3b are quite different from each other, suggesting different occupancy of the two FRET states for different receptors under the same condition - the authors should clarify this in the figure legend and/or in the text. In terms of Gaussian fitting the distributions shown in Fig. 3 and S3, it is not clear which criteria were optimized (chi-squared, AIC, residual correlations, etc); the details provided in Materials and Methods are insufficient, for instance it is not shown how the four peak positions in S3b were chosen (as density plots from HMM analysis weren't shown). The fitted histograms look fine, but with no statistical information provided it is hard to judge the quality of the fitting model.

4. The sequence and docking analysis revealed a conserved loop that is a) longer than in mGluR and b) makes critical contacts stabilizing adjacent protomers in the CaSR dimer. smFRET on the P55L mutant (Fig. 5) showed that the active and inactive states are the same, but the L-Trp shifts the equilibrium to a more open (inactive) state. This is surprising, and suggests that this mutation converts L-Trp from PAM to NAM. I would like to know which molecular-level interactions are responsible for this unexpected effect.

5. The key finding of the paper is the prominent negative charged interface in the LBD2 domain of CaSR and the presumed role in modulating the sensitivity to Ca⁺ activation. The trend in the data shown in Fig. 6 seem to support the idea that electrostatic repulsion is involved in state dynamics. I wonder if the effect of mutations can be further used to described the activation pathway in more detail, i.e., the intermediate state (L-Trp induced); also supporting evidence for the mechanism could be provided from experiments performed in different salt concentrations for counterion screening at the interface between monomers; x-corr analysis may also reveal changes in dynamics associated with these two mutations. Last but not least, more negative charge implies more repulsion, longer distance, lower FRET, yet, the active state is associated with lower not higher FRET.

Response to Reviewer Comments

The reviewer comments are in black, authors' responses are in blue, and how the comments are addressed in the revised manuscript are in yellow highlights.

Overall summary of response to reviewers: We thank the Reviewers for their positive comments on our manuscript, and their insightful questions and providing constructive feedbacks. To address the reviewers' concerns and questions, we have carried out 6 new experiments and analysis and in the revised manuscript we have added 6 main figure panels, 9 supplementary figure panels, and 1 supplementary data file.

We have also significantly revised the manuscript to incorporate reviewers' suggestions. Manuscript line numbers are referenced throughout the point-by-point responses below. 3 reviewers-only figures are also included in this response letter for further clarification.

Figure	Changes
Figure 1	Moved previous panels 1d, 1e to supplementary Fig. 1c,d. New 1b. New panel of published CaSR conformations. New 1e. New analysis showing Ca ²⁺ shifts FRET distribution fully to active state.
Figure 2	Reorganized panels. Updated 2b to include distance measurements.
Figure 3	Updated 3e includes error bars and data points from biological replicates. New 3a. New panel linking FRET change and published structures.
Figure 4	Updated 4d includes error bars and data points from biological replicates. New 4b. Transition density plot for E593UAA.
Figure 5	Added P55L label to 5d.
Figure 6	New 6f. Data from new experiment. New 6g. Data from new experiment.
Figure S1	Rearranged panels to accommodate panels from previous Fig. 1.
Figure S2	Moved previous Fig. S3a to current Fig. S2b. New Fig S2c. Data from new experiment. New Fig S2d. Data from new experiment.
Figure S3	Previous Fig. S3a moved to current Fig. S2b.
Figure S4	New Fig. S4c. Data from new experiment.
Figure S5	Color change in Fig. S5a. New Fig. S5c. New analysis. New Fig. S5d. Structural superposition of CaSR ion-bound structures. New Fig. S5e. Data from new experiment.
Figure S6	Color change in Fig. S6a
Figure S7	No change.
Figure S8 – New	New Fig S8a. New model correlating conformational changes and FRET states. New Fig S8b. Image from smFRET movie. New Fig S8c. Sample traces showing single step photobleaching.
Supplementary Data 1	New supplementary data file to include all the fitting parameters in the data and particle counts.

Reviewer #1

Schamber and Vafabakhsh described the conformational dynamics during CaSR activation using single-molecule FRET experiments. The authors utilize single molecule FRET measurement to monitor the dynamic conformation changes of individual CaSR. They compared two approaches for labeling the CaSR with fluorescent donor and acceptor and conclude that it is more sensitive by inserting unnatural amino acid (UUA) at residue 451 in the CaSR structure and label fluorophore at that location. Then they used this type of smFRET sensor to study the VFT arrangement of CaSR and find out the how negative charge density at the interface of CaSR dimer structure affect its sensitivity to agonist thus the regulation of CaSR activation. Different calcium and amino acid ligand binding conditions in the wild type and several diseases related CaSR mutants were studied. They found that the ligand-free receptor was very dynamic and sampled both active and inactive conformations. Agonist binding stabilized the receptor in the active conformation. Amino acid binding shifted the conformation but did not activate the receptor. By contrast, calcium binding alone significantly increased the occupancy of the active state even though the amino acid was required to fully activate the receptor. The P55L mutation in the LB1 loop caused an increase of the amount of calcium ions required to activate the receptor, consistent with the loss of function of the mutation. High density of negatively charged residues on the LB2 dimer interface was identified and predicted to be a key regulator of CaSR activation. Mutations on the interface either impeded or facilitated the activation depending on whether the mutation strengthened or weakened the negative electrostatic potential. The authors also examined the effect of several disease mutations on the activation process. Although the reported results look interesting and may bring some new insights into CaSR activation, the paper is not ready to publish on Nature Communications at its current stage due to several major issues:

We thank the reviewer for their positive comments about new insights from our study and the constructive comments detailed below.

1) A major claim of the paper is the discovery of the negatively charged LB2 dimer interface modulating the receptor sensitivity. This interface and its role in CaSR activation through electrostatic repulsion have been discussed previously (for example, Zhang et al., 2016, Science Advances 2:e1600241) and so the finding is not completely new.

We thank the reviewer for this comment. We now realize that we have not fully explained the distinction of our proposed model in the manuscript and discussion in the context of existing literature. We do so now in the revised manuscript.

a) The reviewer is correct that the existence of this negatively charged lobe was shown in previous structures. However, molecular mechanism by which it is affecting the activation of CaSR and its role in sensitivity (EC_{50}) tuning of CaSR was not established before. Also, the evolutionary analysis and comparison across class C GPCRs was not done before, to our knowledge.

b) More importantly, we analyzed all currently published CaSR structures that have bound ions (12 structures in total, from 5 independent groups). This analysis is shown below as Fig. 1 for the reviewers and now added as a new Supplemental Fig. 5d in our manuscript. None of the

structures show Ca^{2+} binding at the lower lobe interface and near this negatively charged patch. The structures from the above reference (Zhang et al., 2016, Science Advances 2:e1600241) are the only ones that show any ion binding at or near the negative surface of LB2, but the ions are Mg^{2+} and Gd^{2+} - not Ca^{2+} . Based on the available atomic structures of CaSR bound with Ca^{2+} and our single-molecule results on the full-length receptor, we proposed a model that is essentially different from the model provided in the above reference. Specifically, our new experiments with the salt titration experiments showed that the effect of LB2 can be even Ca^{2+} independent.

Accordingly, to satisfy the reviewers comment we have added the following sentences in our discussion (lines 432-435):

“Finally, while none of the structures show Ca^{2+} binding at negative patch on LB2 (Supplementary Fig. 5d), it is possible that transient or weak binding of Ca^{2+} at this interface can contribute to CaSR activation via screening the negative charges as was previously suggested (Huang, et al., Biochemistry., 2009, & Zhang, et al., 2016, Science Advances).

Fig. 1 for the reviewers – Cartoon representation of twelve active conformation CaSR structures solved in the presence of ions (PDB accession codes: 5K5S, 5FBK, 5FBH, 7M3F, 7M3E, 7DTV, 7DTT, 7E6T, 7DD7, 7DD6, 7DD5, 7M3G) superimposed as viewed from the back of the VFD (a) and rotated 90° degrees to show the dimer interface (b). Calcium (green), magnesium (purple), and gadolinium (blue) ions are shown as spheres. Mutated residues studied in this manuscript R227L, E228K, E249K, E251K, and V258R are colored in orange. Ca^{2+} binding sites are circled and numbered I-V in red. Note that not every structure resolved all of the same Ca^{2+} binding sites.

2) This manuscript also largely overclaimed “We found that several features of CaSR have evolved uniquely compared to mGluRs to satisfy the spatial and temporal constraints of

extracellular Ca²⁺. First, an elongated intermolecular loop linking the LB1s of CaSR and an increased contact area for the LB1 interface likely restrict movement of the ECD in CaSR and holds the LBDs closer to the active-like conformation, resulting in a unique activation mechanism. missed earlier work about probing calcium binding sites”(line 301- 305). This results have also been discovered previously by Zhang et al., (2016, Science Advances 2:e1600241)

This loop was visualized in previous crystal structures, and we have now made this statement more explicit in our manuscript and added the above reference (lines 402-404). However, the referenced paper by Zhang et al. suggested “*involvement of loop 1 in dimerization*” and did not provide mechanistic insights on its role in CaSR activation. In our single-molecule analysis we directly show that mutation of this loop significantly alters the FRET efficiency associated with amino acid binding in CaSR. In the revised manuscript we have also performed smFRET experiments of CaSR with 2 different versions of loop truncation which overall confirm our interpretation. Contrary to the above reference, we did not observe loop truncations have any significant effect on receptor dimerization per se (unpublished results). To address the reviewer’s point, we have added the following in our manuscript (Lines 402-404):

“First, an elongated intermolecular loop previously visualized in crystal structures and involved in linking the LB1s of CaSR (Zhang, et al., 2016, Science Advances.) and an increased contact area for the LB1 interface likely restrict movement of the ECD in CaSR and hold the LBDs closer to the active-like conformation, resulting in a unique activation mechanism.”

2) The authors largely ignored literatures of previously reported calcium binding sites by Yun Huang et al. (JBC, 2005, Biochem 2007) and Zhang et al (2016, Science Advances 2:e1600241). These included a Ca²⁺ binding site with negatively charged residues located at the dimer interface.

It was not the goal of our manuscript to directly identify specific calcium binding sites. The referenced result by the reviewer in Yun Huang et al. (JBC 2007, Biochem 2009) used homology models and mutagenesis to predict potential calcium-binding sites. However, those papers provide no direct evidence for the ability of the predicted binding sites to bind calcium in the full-length CaSR. We want to note that mutagenesis experiments to identify ligand binding sites in GPCRs can be confounded by indirect effects and mutagenesis alone does not prove that a specific position is the binding site. A clear example of this is recent deep mutational scans of GPCRs that show many mutations away from ligand binding or effector binding sites affect efficacy and potency of ligands, without directly affecting ligand-binding (for example Jones, et al., eLife 2021).

The most direct way to identify ligand binding sites is atomic structures. As we discussed in response to a previous comment, we compiled all the available atomic structures of CaSR with Ca²⁺ (Figure 1 for reviewers and new Supplementary Fig. 5d). Our analysis showed that currently 5 Calcium binding sites are represented in available structures with calcium binding sites I-IV present in multiple structures and one structure (PDB accession code: 7E6T) showed binding at site V in the hinge (Fig. 1 for reviewers). Important for this comment, none of the structures show Ca²⁺ binding at the lower lobe or near the negatively charged patch. The only structures that have an ion/metal in that region are from Zhang, et al., 2016, Science Advances, which show Mg²⁺ and Gd³⁺ ions. Even then, the Gd³⁺ ion is not pointed towards the interface and is solvent-facing and

the Mg^{2+} is coordinated by S240 and none of the amino acids we investigated in our study. Therefore, while we cannot rule out the possibility of transient Ca^{2+} binding in the negatively charged patch, currently there is no direct evidence of it in the literature. Nevertheless, we have now acknowledged this possibility in the discussion (Lines 432-435).

We also note that there are currently more than 100 disease-associated mutations annotated in UniProt known to affect CaSR signaling, and many of them are away from any implicated ligand binding sites, further highlighting the extensive allosteric and cooperative nature of activation in CasR. In our assays we have observed several mutations (for example P55L, C129S, and C131G) that affect the EC_{50} of CasR and are away from any possible ligand-binding site. Therefore, we don't think mutational analysis is an accurate reference for ligand binding sites and for that matter in our manuscript we have referenced the multiple available structures as reference whenever needed.

The section describing "Negative charge density of the dimer interface is a key regulator of CaSR activation"(line 207-247) is largely inaccurate due to lack of understanding of the reported role of calcium binding sites.

We have merely stated our findings on this topic. We demonstrated that changing the surface potential of this patch on CaSR changes the occupancy of the active state of the receptor and changes the receptor's EC_{50} correspondingly in a systematic and predictable way.

To further address the concern of the reviewer and strengthen our conclusion, we performed functional experiments on an additional three mutants on the negatively charged patch (E249K, E251K, V258R). This new data is now included in Fig. 6f. These mutants were not previously described in the literature and are away from any ion binding sites in published structures (Fig. 1b for reviewers). The analysis of these mutants is in full agreement with our model, where reducing the negative charge of this patch increases the sensitivity of CaSR for Ca^{2+} .

Finally, our results on mutants R227L and E228K on the negatively charged patch show a change in the relative occupancy of the active FRET state in the presence of 5 mM L-Trp alone (0 mM Ca^{2+}) (Fig. 6d). This result further shows the ability of this charged patch to modulate the relative occupancy of the 0.29 FRET state even in a Ca^{2+} independent context.

As mentioned above, we have now acknowledged the possibility of contribution of this patch to transient Ca^{2+} binding or stabilizing a yet-undefined intermediate state (Lines 432-435).

"Finally, while none of the structures show Ca^{2+} binding at negative patch on LB2 (Supplementary Fig. 5d), it is possible that transient or weak binding of Ca^{2+} at this interface can contribute to CaSR activation via screening the negative charges as was previously suggested (Huang, et al., Biochemistry., 2009, & Zhang, et al., 2016, Science Advances)."

3) The full-length structures of CaSR by Cryo-EM (Ling et al., Cell Research 2021; Gao et al., Nature 2021) have established several signature conformational changes defining the CaSR activation, including the opening/closing motion in the VFT cleft and the LB2 interface, as well as the movement of the LB1 loop. However, none of these locations was chosen to measure the dynamics.

When we started this research in 2017 none of these structures were published, and we used the available crystal structures of the extracellular domain of CaSR to design FRET sensors. We agree with the reviewer that there are several possible potential labeling sites that can be informative. The design, testing, data acquisition, and data analysis for single-molecule FRET is labor intensive, and it is not possible to exhaustively test all potential sites. Additionally, not every position is amenable to mutation or accessible for efficient labeling. For this manuscript, we created and screened 11 Unnatural amino acid (UAA) LB1 sensors, 2 UAA LB2 sensors, 7 UAA CRD sensors, and a half-dozen other constructs that utilized alternative labeling strategies.

The reviewer is correct that a LB1 loop sensor could be informative. Unfortunately, the loop is a technically challenging location for the following reasons: 1) several disease-associated mutations are localized on this loop which is limiting as they affect receptor EC_{50} ; 2) some of residues are not solvent accessible making labelling inefficient 3) Based on the structures, the difference in distance for the loop between inactive and active conformations is quite small and hard to discriminate by smFRET. However, despite all these known issues, during the revision process and based on the reviewer's suggestion we screened S57UAA, E59UAA, and I61UAA as potential candidates because of our desire to be thorough. Unfortunately, none of the three tested locations labeled efficiently enough for single-molecule studies (data not shown) and were not pursued.

The authors need to rationalize their choices of the site to attach the fluorescent probes and confirm that the measured distance changes correlate with the activating conformational changes defined by the Cryo-EM structures.

As suggested by the reviewer we have now added the following rationale for design and verification of FRET sensors (lines 183-184):

"This distance put donor and acceptor probes in the sensitive range for FRET and provide higher spatial resolution than the N-terminal SNAP construct."

The sensitivity and accuracy of distance measurement from smFRET is lower than the CryoEM structures. Moreover, the fluorophores tend to add extra distance, the magnitude of which depends on their orientation and their linkage flexibility. Therefore, direct quantitative comparison of distances from atomic structures and from smFRET measurements is not possible without extra analysis and assumptions. However, crude conversion of FRET values to distances using a generic Förster radius ($R_0 = 54 \text{ \AA}$) yields an approximate distance change of 4.8 \AA between active and inactive states (obtained from smFRET distribution peak centered on 0.41 and 0.29). To further address the reviewer's comment regarding the comparison between our results and the Cryo-EM structures, we have now added a new figure to specify the direct mapping between structures and smFRET measurements (Fig. 2b and 3a). We have also edited lines 188-190:

“This is again consistent with our previous result of a small VFT domain conformational rearrangement upon activation and published structures that show 3.6 Å difference between the Ioo and Acc conformations (Fig. 2b).”

4) What new here is probably the sequence comparison across homologs. However, this analysis seems to be too simple to establish the evolutionary role of the interface. A more quantitative evaluation on the electrostatic potentials of CaSR homologs and their relationship to the receptor sensitivity is required to validate the authors' prediction.

We appreciate that the reviewer did not identify a conceptual or technical shortcoming with our analysis. We originally pursued the homology sequence analysis to ask if the negative patch is evolutionarily conserved. Once the analysis showed that this patch is conserved, we developed further analysis to quantify the variation in this region across evolution. To put our analysis in a context, we compared the results to other parts of the protein and to a closely related protein mGluR1. While our presented analysis is conceptually simple, it is unbiased and does not use any assumption or parameters and importantly includes internal normalization and controls. We think the simplicity makes it reliable and easy to understand, which we consider to be a strength of the analysis.

To address the generality of a fundamental relationship between charge and Ca²⁺ sensitivity and to address the reviewers concerns we are including two new sets of experiments:

4.1- We have now included functional data on three additional mutations (E249K, E251K, V258R). All these mutations are novel and were not previously described (verified via GPCRdb & UniProt). All these mutations were designed to simply introduce positively charged residues into the LB2 surface to test this hypothesis. All these mutations increased Ca²⁺ sensitivity and decrease EC₅₀ of CaSR for Ca²⁺. We have now included this data as a new Fig. 6f. these three novel mutations (E249K, E251K, V258R) combined with the two previously analyzed mutations in the manuscript (inactivating: R227L; activating: E228K) further support our model on the role of charged residue distribution at the LB2 in modulating calcium sensitivity.

4.2- To further explore the role of electrostatic repulsion in regulating the relative occupancy of the activate FRET state, we performed smFRET experiments where we titrated increasing concentrations of NaCl in the absence and presence of 10 mM L-Trp. There was no discernible change in the FRET distribution in the absence of CaSR ligands, but in the presence of 10 mM L-Trp alone, increasing the NaCl concentration resulted in a progressive shift towards the lower FRET active state. These new data provide additional support for the role of negative charge in our model, independent of Ca²⁺. This data has been included as a new Fig. 6g.

5) While the smFRET imaging experiments are quite interesting and useful, the authors need to address and clarify following questions:

a. Do you have further evidence support your hypothesis that the difference in the distribution of intermolecular interaction at LB1 is responsible for effect of amino acid binding in CaSR and mGluRs? (For example, with mutations which increase the intermolecular interaction at LB1, mGluRs response to ligand binding in a way more similar to CaSR, or vice versa.)

We thank the reviewer for asking this question and giving us the opportunity to discuss this point better. This is an important, but difficult question to answer because it is still technically challenging to predict an effect of a mutation on structure and dynamics. We agree with the author that it is an exciting thought to convert mGluR to a CaSR-like receptor (or vice versa) and is something we are actively investigating.

An indirect support of our model is the recent cryoEM structure of inactive CaSR in complex with a nanobody (Chen, et al., 2021, eLife). In this structure the nanobody pushes the receptor into a more mGluR-like inactive state as seen by an increased distance between LB2-LB2 (Fig. 2a for reviewers). Importantly, in this CaSR-nanobody complex the interprotomer loop is pointing away from the adjacent protomer (Fig. 2a for the reviewer) suggesting that the contacts made by the interprotomer loop are not compatible this mGluR-like conformation. Notably, this structure is the only structure where the loop is partially disordered (Fig. 2a for the reviewer, right). Notably with regard to this comment, the contact area between the protomers also shows a reduced footprint that is more similar to mGluR1 (Fig. 2b for the reviewer).

This structure is consistent with our interpretation that the increased surface area observed in CaSR compared to mGluR1 is attributed to interactions made by this interprotomer loop and results in a LB2-LB2 distance that is smaller than the mGluRs.

We have performed new experiments comparing the ability of L-Trp binding to modulate the response of wild-type and mutant CaSR. After creating a loop truncation mutant ($\Delta 47-57$), we observed the addition of L-Trp resulted in a 40%, 45%, and 58% reduction of Ca^{2+} EC_{50} for wild-type, P55L, and $\Delta 47-57$ respectively. These new data support our claim that the long interprotomer loop in CaSR reduces the contribution of amino acid binding in activation and is a primary difference between CaSR and mGluRs. These data are now included in a new Supplementary Fig. 4c.

Fig. 2 for the reviewers – **a** Surface and ribbon representation of the Inactive (open-open) cryoEM structure (left) and the Inactive (open-open, in the presence of a nanobody) cryoEM structure (right). Interprotomer loop residues 40-54 colored in gold and residues 55-67 colored in blue to show relative orientation of the loop (black arrows). **b** Surface representations of inactive CaSR in the absence of ligand (left), inactive CaSR when complexed with nanobody (middle), and inactive mGluR1 in the absence of ligand (right). Residues that contact the adjacent protomer colored in red.

b. Do you have explanation about: during activation, how can the ligand binding overcome the electrostatic repulsion of the LB2 domains? Especially why binding L-Trp can bring the state from inactive towards active (not fully active).

It is possible that the new conformation of ECD after ligand binding provide new interactions that can stabilize the active state. Moreover, direct interaction between receptor domains beyond ECD that form in the active CaSR, such as between transmembrane helices (for example TM6-TM6

with possible contribution from membrane cholesterol) and CRD–ECL2–ECL3 network, can help stabilize the active conformation. Our methodology and data cannot answer this question for sure and therefore we have avoided making claims with this regard. However, in addressing the reviewer’s previous comments, we have made changes that attempt to address this question where we emphasize the congruency between our data and the established model of counterion screening by Ca^{2+} . (Lines 432-435)

c. Is the 30ms temporal resolution sufficient to revolve the dynamic transitions between different conformations of CaSR?

There could be dynamics that is faster. However, we have repeated some of the key experiments with 5 ms (200 Hz) data acquisition, which is the fastest currently possible data acquisition rate with EMCCD camera-based smFRET experiments and obtained similar results. we have added the following sentence in the discussion (lines 421-422) to clarify this important point.

“Finally, it is possible additional intermediate states exist that we did not resolve due to the spatial and temporal limitations of smFRET measurements.”

d. The authors should provide single molecule images and movies of both donor and acceptor signal in their experiments.

We now included a new Supplementary Fig. 8b showing images of donor and acceptor channels. At the suggestion of the reviewer we have uploaded a sample movie for D451UAA in the following conditions: 3 mM EDTA, 5 mM L-Trp, 10 mM Ca, 10 mM Ca + 5 mM L-Trp to the Harvard dataverse (<https://doi.org/10.7910/DVN/PKR9SD>). The database allows for 10 Gb per dataset, and these 4 movies meet that limit. Making the complete dataset for this manuscript available (approximately 8TB) is difficult due to the size limitations of available options. While we are exploring other services to upload larger datasets, as mentioned in our data availability statement, we will also accommodate direct requests for data access.

e. How to prove your FRET is single molecule? Usually, this can be simply done by photobleaching experiments. The authors added an oxygen scavenger system to minimize the photobleaching effects. However, by imaging for sufficient long time, e.g. tens of minutes, one should be able to eventually photobleach the single molecule fluorescence. The author needs to provide such information in their Supporting information.

We thank the reviewer for the question. We now include a new Supplemental Fig. 8c with examples of single molecule bleaching to illustrate our statements in lines (801-805):

“Only particles that showed a single donor and a single acceptor bleaching step during the acquisition time (Supplementary Fig. 8c), stable total intensity (ID + IA), anticorrelated donor and acceptor intensity behavior without blinking events and that lasted for more than 3 s were manually selected for further analysis...”

f. Usually, one can acquire very large data set for single molecule imaging. For most of the experiment data in this paper, 3 individual experiments were carried out. It is not clear what exactly

does the '3 individual experiments' stands for? Does it mean 3 individual single molecule measurements? Or does it mean 3 trials with many single molecule measurements each time? If only 3 single molecule measurements are accomplished, the authors definitely needs to acquire more data. If it is 3 individual trials, the authors need to make it clear in their statement.

We thank the reviewer for bringing up these points so we can clarify our meaning. By '3 individual experiments', we meant '3 individual biological replicates' and have adjusted the manuscript to reflect this (Many lines throughout the manuscript).

To reduce confusion, we have also made the following changes to our methods section (line 811):

"A minimum total of 300 FRET traces from 3 individual biological replicates were used to generate population smFRET histograms unless otherwise stated. "

To address question about the number of single molecule measurements in our data, we have included a table of particle counts for each biological replicate in a new supplementary file.

Minor concerns:

1. Missing error bars in Fig. 3d, 4c, 6d. How many experiments did? Need to put error bar.

Figs. 3d, 4c, or 6d are the fitted area under the curve for the averaged FRET histograms. These histograms are themselves an average of 3 biological replicates. Based on the reviewers' feedback, we performed peak fitting on the individual biological replicates and have updated the figures to include error bars and data points (Now Figs. 3e, 4d, & 6d).

We have also updated the legends for the figures to reflect this change:

Fig. 3e Occupancy of the two FRET states of the VFT in the presence of increasing ligand concentrations. Values represent mean \pm s.e.m. area under individual FRET peaks centered at 0.41 (inactive) and 0.29 (active) from smFRET population histograms. Data represent mean \pm s.e.m. of at least N = 3 independent biological replicates.

Fig. 4d Occupancy of the four FRET states of the CRD in the presence of increasing ligand concentrations. Values represent mean \pm s.e.m. area under individual FRET peaks centered at 0.22 (inactive), 0.38, 0.56, and 0.78 (active) from smFRET population histograms. Data represent mean \pm s.e.m. of at least N = 3 independent biological replicates.

Fig. 6d Occupancy of the active FRET state of R227L (red) and E228K (green) for each condition normalized to wild-type. Values represent mean \pm s.e.m. area under active FRET peaks from smFRET population histograms, averaged over 3 independent biological replicates, centered at 0.41 (inactive) and 0.29 (active).

Can the authors explain more about Fig.6d? Why there is no bar for Occupancy of E228K at 40mM Calcium + 5mM L-Trp?

There is no bar for E228K occupancy at 40 mM Calcium + 5 mM L-Trp because that condition was not used for E228K, as 10 mM Calcium + 5 mM L-Trp was sufficient to fully activate the receptor (Fig. 6c). Since the R227L mutation reduces calcium sensitivity, additional calcium was needed to fully shift the histogram and reach saturation. To clarify this, we have added the following sentence to the Fig. 6d legend (Lines 576-577):

“40 mM Ca²⁺ + 5 mM L-Trp condition was not tested for E228K.”

For studying the CRD of CaSR, four conformation states are used to describe the FRET distribution (Fig. 4). How do the authors determine the peak positions in their fitting here?

We used the transition density plot to identify the number and position of the peaks. We have now included a new panel (Fig. 4b) that shows the transition density plot and included in the figure legend how it is used as the rationale for the peak positions.

(Lines 541-542) “Dashed lines represent the most frequently observed transitions and were used for multiple-peak fitting of FRET histograms.”

We have also updated our methods section to include this clarification as well (Lines 821-824):

“Peak centers (xc) were constrained as mean FRET efficiency of a conformational state ± 0.02 (2x histogram bin size). The mean FRET efficiencies associated with different conformational states was determined based on the most frequent transitions between FRET efficiencies observed in transition density plots, which are denoted by dashed lines (Fig. 2f, 3d, 4b).”

For Supplementary Fig. 1a, what cell line do you use?

The figure legend for Supplementary Fig. 1a and the methods section titled “Calcium Mobilization Assay” have been updated to include cell line information of “HEK293T cells” in lines 610 & 842:

Response profile of HEK293T cells expressing SNAP CaSR (top).

Coverslips with HEK293T cells expressing N-terminal SNAP CaSR were briefly washed in...

Does this cell line express endogenous CaSR? Do you have results of negative (cell with empty vector transfection) and positive (cells transfected with nonmodified WT CaSR) control?

According to the Human Protein Atlas HEK293 cells do not express CaSR (<https://www.proteinatlas.org/ENSG00000036828-CASR/cell+line>). We have verified this as well by performing experiments with untransfected cells, empty plasmid, or mGluR2 plasmid.

When you say cell imaging, did you monitor single cells or a cell population? How many cells does this N=3 individual experiments include?

We have updated the manuscript to clarify how our cell imaging data was analyzed:

(Lines 859-861) “Movies were analyzed in Fiji by manually drawing a region of interest (ROI) centered on individual cells that showed labeling with Alexa-647 dye (50 to 100 cells per field of view resulting in a minimum of 150 cells across 3 individual biological replicates) indicating cells expressing CaSR. Cells without labeling did not respond to changes extracellular calcium.”

(Lines 867-869) “Response profiles of individual cells were summed and treated as a single ROI before quantification of response and fitting of a dose-response curve.”

Reviewer #2:

In the manuscript ‘Mechanism of Sensitivity Modulation in a Class C GPCR via Electrostatic Tuning’, Schamber et al. used single-molecule FRET, sequence analysis and signaling assay to study conformational changes in the calcium-sensing receptor. Specifically, the authors mapped the ligand-independent and ligand-dependent conformational dynamics of the CaSR N-terminal domain. In addition, they identified a negatively charged patch at the dimer interface of CaSR that is involved in fine-tuning the receptor sensitivity toward extracellular Ca^{2+} . This study provides new information on how conformational changes propagate along the receptor. Their identification of the structural hub that allosterically controls receptor activity through electrostatic repulsion is an advance in the field.

We thank this reviewer for their detailed reading, and their positive comments.

Here are some issues that need to be addressed before publication.

Major:

1. The authors concluded that amino acids fit the role of allosteric modulators because L-Trp increases transitions between the active and inactive states, but these transitions are brief and do not increase the active state occupancy sufficiently to result in signaling output. On the other hand, 10mM Ca^{2+} alone was able to significantly increase the occupancy of the active state. Is it possible that some amino acids or amino acid analogs are already bound to the receptor and are cooperating with the added Ca^{2+} to improve the occupancy of the active state?

Previous structural studies of the CaSR (Ling et al., 2021; Chen et al., 2021; Gao et al., 2021; Geng et al., 2016; Zhang et al., 2016) all indicated the presence of an amino acid-like compound in Ca^{2+} or Mg^{2+} -bound CaS receptor structures. This compound has been identified to be a Trp derivative (Zhang et al., 2016), and it has sufficient affinity for CaSR that it remains bound to the receptor throughout the protein purification steps.

We believe that this is an important question. We are aware of the potential for residual amino acid or their derivatives binding in the cleft as reported in some structural studies. For our smFRET assays, after immobilization of receptors at ~ 50 pM CaSR concentration, we wash the flow chamber with 50x chamber volume. However, due to the importance of this question, we performed additional controls to confirm the ability to efficiently remove amino acids by washing. First, we determined the ability of possible bound L-Trp to be removed by washes in our experiments. We collected smFRET datasets before, during, and after the presence of 5 mM L-Trp. We found that shift in the FRET distribution in the presence of 5 mM L-Trp is reversible under typical wash conditions in our experiments (Supplementary Fig. 2d) suggesting we can remove bound amino acids by washing our flow chamber. Next, to test whether amino acids or their derivatives remain bound to immobilized receptor, we acquired a dataset before we performed an

extended wash over 2 hours. Every 30 minutes 50x chamber volume of wash buffer was passed through the flow chamber. After the fifth wash at 2 hours, we acquired another dataset. The resulting FRET distribution in this experiment is like the reported data in the manuscript and the dataset pre-wash. This result is consistent with the interpretation that there are no residual ligands bound to the cleft of CaSR's VFT. After this extended wash to verify the absence of residual amino acids, we observed a shift towards lower FRET upon the addition of 10 mM Ca²⁺ alone. These data are included in new panels Supplementary Fig. 2c and including the following statement to Results (Lines 230-239):

“To ensure that our observation is not due to residual L-Trp bound to CaSR during the purification (add refs to those studies that show this), we performed an extended wash of immobilized receptor and failed to detect any change in the FRET distribution post-wash while the distribution again shifted to lower FRET in the presence of Ca²⁺ alone (Supplemental Fig. 2c). To further determine if amino acids could be removed in our experiment, we collected smFRET data before, during, and after the presence of 5 mM L-Trp. We found that 5 mM L-Trp reversibly shifted the FRET histogram (Supplemental Fig. 2d) suggesting we can remove amino acids bound to immobilized receptors. Based on these data, it is unlikely that the effect observed in the presence Ca²⁺ alone to be caused by residual amino acid binding. Together, these observations are consistent with the role of amino acids as allosteric modulators^{31,36}.”

We have added the following to the methods section explaining the extended wash protocol (Lines 787-790):

“For the extended wash, a dataset was acquired prior to passing 50x chamber volume of wash buffer through the flow chamber every 30 minutes for 2 hours. After the fifth wash at 2 hours, a dataset was acquired to compare to the pre-wash data. Finally, a dataset was acquired in the presence of 10 mM Ca²⁺ and 10 mM Ca²⁺ + 5 mM L-Trp.”

Fig. 3d indicates that L-Trp alone is able to induce or increase the presence of active conformation. The authors also showed that only the combination of 10mM Ca²⁺ and 5mM L-Trp fully shifts the occupancy of the active state. These data seem to argue that the amino acids are partial agonists, or co-agonists with Ca²⁺, not allosteric modulators. This would also be in agreement with findings from the earlier structural studies that amino acids bind to the agonist site in the VFT (Ling et al., 2021; Chen et al., 2021; Gao et al., 2021; Geng et al., 2016; Zhang et al., 2016).

Whether L-Trp is an allosteric modulator of CaSR or a co-agonist has been subject of several previous publication with existing evidence for both models in the literature. For example, previous work has shown the ability to delineate the modulatory effect of amino acid binding from the effect of Ca²⁺ via mutagenesis (Mun et al., 2004, Journal of Biological Chemistry, <https://doi.org/10.1074/jbc.M500002200>) suggesting that L-amino acid binding is not required for CaSR activation. A recent spectroscopic study showed that Ca²⁺ alone was able to activate CaSR following dialysis to remove possible residual L-amino acids (Liu, et al., 2020, PNAS, <https://doi.org/10.1073/pnas.1922231117>).

The question of whether amino acids function as co-agonist is centered on whether their binding is necessary (or not) to achieve CaSR activation. In addressing the reviewer's previous comment,

we have demonstrated the reversibility of L-Trp binding and detected no change in the FRET distribution after a 2-hour extended wash. Because of these data, we conclude there are no amino acids or their derivatives remaining bound to immobilized CaSR in our assay.

For determining the ability of a ligand to activate CaSR, the E593UAA sensor is a better reporter of activation because it reports on conformational rearrangement of the CRD and is closer to the G-protein binding pocket than the D451UAA or SNAP sensors. For this sensor (E593UAA), our data show that 5 mM L-Trp induces little to no change in occupancy of the 0.78 FRET state when compared to the 3 mM EDTA condition (Fig. 4d), and this result is consistent with what is observed in the FRET distributions (Fig. 4c). This contrasts with the 4 mM and 20 mM Ca²⁺ conditions where there is an increase in occupancy (Fig. 4d) and a noticeable shift in the FRET distributions (Supplementary Fig. 3c). Because of the effect of Ca²⁺ in the absence of L-Trp, we conclude that L-Trp (and amino acids) are not necessary for CaSR activation.

While the reviewer is correct that 10 mM Ca²⁺ and 5 mM L-Trp shift the SNAP-CaSR and D451UAA sensors fully to their respective 0.21 and 0.29 FRET peaks, we have included a new Fig 1e that shows Ca²⁺ has the same effect at a sufficiently high concentration.

We have added the following sentences referencing this panel (lines 165-166):

“Calcium alone was able to fully shift the FRET distribution to the FRET state corresponding to the Acc conformation (Fig. 1e).”

2. The authors stated amino acids facilitate VFT rearrangement. Are the authors referring to VFT closure? Does the transient FRET transition induced by L-Trp correspond to a transient LB1-LB2 closure? Are the authors implying that the VFT remains open majority of the time even when L-Trp is bound?

We agree with the reviewer that our language was confusing. We have included several changes to address this.

We have included new figures to provide a clearer mapping between our data and the structures and what we mean by VFT rearrangement (Fig. 3a and Supplementary Fig. 8a).

We have edited the manuscript to specify more clearly what our data shows:

lines (207-215) “Published cryoEM structures show a distance change of -0.7 \AA measured at the D451 C α as the receptor transitions from the I_{oo} conformation to the I_{cc} conformation upon L-Trp binding (Fig. 3a). This change is below the resolution of smFRET and therefore these two states should appear as a single peak in our smFRET measurement. However, we observed that 2.5 mM L-Trp alone resulted in a smFRET peak centered between the active and inactive FRET states (Fig. 3b). This implies that amino acid binding is causing a change in conformation beyond that shown in the I_{cc} structure, and the unexpected change in FRET could be caused by the stabilization of a novel conformation or induction of rapid exchange between the 0.29 and 0.41 FRET states.”

(lines 223-225) “Therefore, these results are consistent with a model that amino acids not only induce closure of the VFT as observed in structures, but also increase the occupancy of the 0.29 FRET state suggesting transient engagement of the LB2-LB2 interface.”

(lines 353-354) “The ability of NaCl to modulate the effect of L-Trp alone is consistent with our interpretation that amino acid binding induces a partial or transient engagement of the LB2-LB2 interface.”

We have also included the following to the discussion section (Lines 415-420):

“We observed the ability of L-Trp to induce an increase in occupancy of the 0.29 FRET active state for D451UAA, but we did not observe a similar increase in the occupancy of the 0.78 FRET active state for E593UAA which probes the CRD. Because of this, we are not able to conclude that amino acid binding induces the Acc conformation characterized by engagement of both the LB2-LB2 interface and the CRD-CRD interface (Supplementary Fig 8a).”

We have also adjusted the title of the section to be more precise (line 199):

“Amino acids facilitate VFT rearrangement beyond VFT closure.”

3. Fig. 6e shows that the charge distribution on the surface of LB2 varies across species. The authors stated that this variation is correlated with calcium sensitivity of CaSR. Are the authors implying that low negative density on LB2 results in higher sensitivity to Ca^{2+} (EC_{50}) in some organisms and vice versa? Are there any functional data that support this kind of hypothesis?

Our data showed that if we increase or decrease the negative charges on this specific patch at the dimeric interface of CaSR, the receptor EC_{50} systematically increase or decreases. We showed this result with 2 mutations in our original submission, and we have now added 3 additional mutations: E249K, E251K, and V258R. These data have been included as a new Fig. 6f.

To compare between organisms, we generated structural homology models of CaSR orthologs that have EC_{50} data published (goldfish and Atlantic salmon) using the SWISS-MODEL server and crystal structure (5K5T) as the template structure. We performed the charge density analysis following the same protocol as described in the manuscript. The surface of both homologs have relatively more positive surface charge compared to human CaSR, and based on literature, they both have lower EC_{50} than the human CaSR (Fig. 3a,b for reviewers). One caveat of this analysis is that the functional data for EC_{50} of CaSR of different species are not acquired in the same buffers. We could not obtain these plasmids within a reasonable time to test them ourselves and our efforts to clone part of CaSR from these organisms into the human CaSR was also not successful.

Fig. 3 for reviewers – **a** Surface representation of human, goldfish, and atlantic salmon CaSR colored by electrostatic potential where red is negative and blue is positive. **b** Multiple sequence alignment for human, goldfish, and atlantic salmon for the LB2 Helix-Sheet-Helix structural motif with positively charged residues highlighted in blue, and the negatively charged residues highlighted in red.

Minor:

1. Line 33. Not all the ECDs of Class C GPCRs are covalently linked. An example is the heterodimeric GABAB receptor.

We thank the reviewer for this comment, and we have updated the manuscript accordingly as following (Line72):

“Among them, class C GPCRs are constitutive dimers with ~600 amino acid extracellular domain that, in some members, are covalently linked^{6,7} (Fig. 1a)”

2. Fig. 3a. It is difficult to distinguish the curves for 5mM Trp and 10mM Trp. Are they overlapping?

The curves for 5 mM L-Trp and 10 mM L-Trp are overlapping, and we have added a note in the to clarify this in the figure legend (Line 527):

“Histograms for 5 mM L-Trp and 10 mM L-Trp are overlap.”

3. Is Fig. 3b. Both panels are labeled 5mM L-Trp. Is this correct?

Both panels are 5 mM L-Trp. We wanted to highlight the heterogeneity in single-molecule experiments and show both the brevity of transitions to the active state as well as effect of L-Trp on the frequency of transitions. We see how our writing could be confusing and we have modified language to clarify this as following (Line 530-531):

"Sample traces show particles exhibiting different behaviors in the same condition with infrequent and very brief transitions (1-2 datapoints), or frequent and brief transitions (5-10 data points)"

4. Please indicate the construct used to generate Fig. 5d both in the figure and in the legend.

We have now included a label in Fig. 5d and legend clarifying that the construct is P55L (Line 556).

"d smFRET population histogram for P55L in the presence of ..."

Reviewer #3:

This study maps the millisecond-scale conformational states and dynamics of the extracellular domain of the calcium sensor CaSr, a dimeric class C GPCR, via TIRF smFRET. The single-molecule data is corroborated with signalling assays, protein sequence and mutation analysis to infer mechanistic details about inter-protomer interactions, in particular the role of electrostatic repulsion/attraction in tuning the activation of CaSR receptor. Overall this is a well designed and carried out study and the model proposed by the authors is supported by the experimental evidence. However, I have some questions/concerns about the data analysis and the interpretation:

We thank this reviewer for their positive comments.

1. Fig. 1 data and description on page 4; A FRET shift of 0.04 under activating conditions is attributed exclusively to a distance change of 2.5 Å, but no indication is given about error margins as well as other factors, such as the orientational factor in FRET.

We agree that discussion of FRET error and factors that contribute to the error is important to any robust quantitative discussion of smFRET. We did a generic distance conversion for qualitative comparison with the published structures. Because our goal was not to quantitatively measure distances, we have not included error margins or other factors like orientation factor. We have now made the qualitative nature of the smFRET to distance conversion clearer throughout the manuscript. As we discussed in a response to a previous comment, we have also added a new figure to show direct mapping between our smFRET states and the structures (Fig. 3a).

Furthermore, the cross-correlation analysis (1d) lacks numbers/lifetimes to support statement like "the receptor is very dynamic in the absence of ligand" and that this was reduced in the presence

of agonists. Overall, tables with fitting parameters/errors would serve the authors well to drive their points across.

We thank the reviewer for raising this important point. As per the suggestion, we have included a spreadsheet as supplementary table (Supplementary Table 1) that includes parameters of all fits. We have also updated our language as follows to indicate the metric we use to justify statements regarding dynamics:

(Line 157-159) “Quantification of receptor dynamics by cross-correlation analysis showed that the receptor is most dynamic in the absence of any ligand compared to conditions with ligands as quantified by cross-correlation amplitude (Supplementary Fig. 1c, Supplementary Data 1)”

(Lines 163-164) “In the presence of agonists, the amplitude of the cross-correlation was reduced (Supplementary Fig. 1c) ...”

About x-corr analysis, controls for "slow" vs. "fast" state exchange would be very helpful. Similarly, static/dynamic controls for FRET width analysis (1e) would also be beneficial, especially when drawing conclusions about the 5-ms data presented in Fig. S1.

There are multiple factors that affect the amplitude and timescale of cross-correlation including frequency of transitions, lifetime of states, FRET values and Δ FRET between states, and noise. Without detailed kinetic experimental data to limit the parameter space there can be many combinations of parameters that can create qualitatively similar outputs.

The point we were trying to make was the stark difference in the receptor dynamics between CaSR and mGluR2. For CaSR we observed many transitions that lasted less than 2 datapoints, even when we acquired data at 200 Hz. In our revision we tried to be clearer about direct comparison between mGluR and CaSR.

2. Fig. 2 data and description; the D451UAA sensor is indeed more sensitive to FRET changes (0.12 vs. 0.04), which raises the issue why data on the previous construct (Fig. 1) should still be included in the paper.

We left the SNAP-CaSR construct in the paper for three reasons: 1) The results of the SNAP construct match the results of the UAA construct qualitatively and serves as a good control. 2) Several previous experiments were done with N-terminal SNAP-mGluR2 constructs, in the same format as the SNAP-CaSR here. This allows a direct comparison between the two receptors. 3) Finally, for our study of mutants (Fig. 7b,c), we found that the combination of the mutation D451UAA and some disease-associated mutations greatly affected the trafficking and expression of the receptor, which hindered study by single-molecule FRET. For those experiments we used the SNAP-CaSR construct. In consideration of the reviewer's suggestion to better emphasize D451UAA, we have rearranged panels between Figures 1 and 2 as well as Supplementary Figures 1 and 2.

It is not clear how the dynamics inferred from this construct (x-corr lifetimes) compare to previous construct (Fig. 1 and S1), especially at 5-ms resolution.

When comparing the SNAP and D451UAA sensors, we were referring to the reduction of the cross-correlation amplitude upon addition of ligands which we interpret as the density of dynamics. The relative ordering of the cross-correlation amplitude across the conditions tested in this study are consistent between the SNAP and D451UAA FRET sensors. We have modified the language of lines (190-192) to more clearly express this:

“Like the N-terminal SNAP sensor, addition of ligands reduced the cross-correlation amplitude of D451UAA (Fig. 2d). “

In addition, it is not clear whether/how panel f density plots were (or can be) used to infer the populations of different conformational states of CaSR in different ligand conditions.

We have made the following changes to figure legends to clarify how the transition density plots were used to determine the fret efficiency values associated with different conformational states:

(Lines 517-518, 533-534, 541-542) “Dashed lines represent the most frequently observed transitions and were used for multiple-peak fitting of FRET histograms.”

We have also updated our methods section to include this clarification as well (Lines 821-824):

“Peak centers (x_c) were constrained as mean FRET efficiency of a conformational state ± 0.02 (2x histogram bin size). The mean FRET efficiencies associated with different conformational states was determined based on the most frequent transitions between FRET efficiencies observed in transition density plots, which are denoted by dashed lines (Fig. 2f, 3d, 4b).”

3. Fig.3 data (L-Trp titration); the examples shown in 3b are quite different from each other, suggesting different occupancy of the two FRET states for different receptors under the same condition - the authors should clarify this in the figure legend and/or in the text.

We thank the reviewer for this suggestion, and we have edited the legend of Fig. 3b to clarify the different behaviors of the sample particles (lines 530-532).

“Sample traces show particles exhibiting different behaviors in the same condition with infrequent and very brief transitions (1-2 datapoints), or frequent and brief transitions (5-10 data points)”

In terms of Gaussian fitting the distributions shown in Fig. 3 and S3, it is not clear which criteria were optimized (chi-squared, AIC, residual correlations, etc); the details provided in Materials and Methods are insufficient, for instance it is not shown how the four peak positions in S3b were chosen (as density plots from HMM analysis weren't shown). The fitted histograms look fine, but with no statistical information provided it is hard to judge the quality of the fitting model.

We agree with the reviewer's comments, and have made the following changes: We have added the following to the methods section titled “smFRET data analysis” to specify the optimization metric (lines 824-826):

“Peak fitting used the LevenBerg-Marquardt algorithm to determine best fit by Chi-square with a tolerance of 1E-9 in OriginPro.”

We have included a new Fig 4b to show the TDP that was used to Justify the position of the 4-peaks. In response to reviewer comment 3.3 above, we have included a supplementary spreadsheet containing all relevant parameters from fitting.

4. The sequence and docking analysis revealed a conserved loop that is a) longer than in mGluR and b) makes critical contacts stabilizing adjacent protomers in the CaSR dimer. smFRET on the P55L mutant (Fig. 5) showed that the active and inactive states are the same, but the L-Trp shifts the equilibrium to a more open (inactive) state. This is surprising, and suggests that this mutation converts L-Trp from PAM to NAM. I would like to know which molecular-level interactions are responsible for this unexpected effect.

We agree with the reviewer that the effect of L-Trp on the P55L mutant is a surprising result. To carefully investigate the physiological effect of this mutation and how it is affected by L-Trp, we performed titration experiments and measured the EC_{50} . P55L mutant has an EC_{50} of 9.5 mM in the absence of L-Trp and an EC_{50} of 5.2 mM in the presence of 5 mM L-Trp. Therefore, L-Trp acts as a PAM for the P55L mutant. We have included a new Supplementary Fig. 4c to report this data.

5. The key finding of the paper is the prominent negative charged interface in the LBD2 domain of CaSR and the presumed role in modulating the sensitivity to Ca^{+} activation. The trend in the data shown in Fig. 6 seem to support the idea that electrostatic repulsion is involved in state dynamics. I wonder if the effect of mutations can be further used to described the activation pathway in more detail, i.e., the intermediate state (L-Trp induced);

Based on our data and sensitivity of smFRET measurements and available structures we think the simplest model to fit our data is the two-state model we presented. Within the temporal and spatial sensitivity limitations of our measurement, the mutants can be also explained in the same framework. However, we cannot rule out the possibility of a short-lived intermediate state or a conformational state with small distance changes which are beyond spatial and temporal sensitivity of our measurement.

We have added a figure to address another reviewer’s comment that can clarify mapping between our smFRET data and the structures (Fig. 3a and Supplementary Fig. 8a).

also supporting evidence for the mechanism could be provided from experiments performed in different salt concentrations for counterion screening at the interface between monomers;

We thank the reviewer for this interesting suggestion. We performed this NaCl titration smFRET experiment in the presence and absence of L-Trp and we have now included the results as a new Fig. 6g.

These experiments indeed showed that the smFRET distribution shifted further towards the active FRET state in the presence of 5 mM L-Trp alone when the concentration of NaCl increased (Fig. 6g) further suggesting the role of electrostatic repulsion in regulating active state occupancy.

We have also included the following statement to the manuscript:

"We found that FRET distribution shifted further in the presence of 5 mM L-Trp when the concentration of NaCl increased (Fig. 6g and Supplementary Fig. 5e) further suggesting the role of electrostatic repulsion in regulating occupancy of the 0.29 FRET state. The ability of NaCl to modulate the effect of L-Trp alone in the absence of Ca^{2+} is consistent with our interpretation that amino acid binding induces a partial or transient engagement of the LB2-LB2 interface."

x-corr analysis may also reveal changes in dynamics associated with these two mutations.

We appreciate the reviewer's suggestion, and we have included cross-correlation plots for the two mutants as a new Supplementary Fig 5c.

Last but not least, more negative charge implies more repulsion, longer distance, lower FRET, yet, the active state is associated with lower not higher FRET.

The D451UAA sensor reports movement at the top of the flytrap in the upper lobe and not the distance between the lower lobes. Consistent with the published structures, the residue D451 is closer in the inactive conformation than in the active conformation, which results in lower FRET being associated with the active conformation. We have added Fig. 3a that shows the measured distance between C α of D451 in several structures to help clarify.

REVIEWERS' COMMENTS

Reviewer #1 (Remarks to the Author):

In this revision, Schamber and Vafabakhsh have responded to the majority of the previously concerns with additional data and revision. The following additional revisions are suggested:

1). Major conclusions are based on the assumption that smFRET constructs accurately reflect the cellular CaSR in live cell activation processes. The smFRET signals were achieved using SNAP-tag with labeling and dyes in a fixed cell lysate measurement, or were based on non-natural amino acid UAA substitution at either position D451 (D451UAA0), or at E593 (E593UAA) with dye labeling. Please add dose response curves for these constructs and compare with w.t without labeling, using live cells with calcium imaging, to fully justify that “these constructs are active with EC50 comparable to wildtype receptor” (line 146-148), since CaSR’s EC50 changes and can depend on experimental buffer conditions.

2) Please modify the following sentence: “Finally, while none of the structures show Ca²⁺ binding at the negative patch on LB2 (Supplementary Fig. 5d), it is possible that transient or weak binding of Ca²⁺ at this interface can contribute to CaSR activation by screening the negative charges, as suggested in a previous study (Huang, et al., Biochemistry., 2009, & Zhang, et al., 2016, Science Advances) Line 433-434)”. A calcium binding site was reportedly involved in E228 and E229 at the negatively charged surface between lobe 2 (LB2) by Huang et al. (Biochem 2007), based on purified protein fragments, mutational studies (E228I, E229I), and calcium response functional assays. In previously reported studies by Huang et al. 2007 and Zhang et al, 2016, removal of calcium binding ligand residues E228I and/or E229I significantly increased the EC50 of intracellular calcium responses of this receptor, supporting the role of calcium binding at this negatively charged patch location in the activation of the dimer form. These previously reported results are different from the statement in this manuscript that “the regulatory effect of negatively charged LB2 patch is independent of Ca²⁺” (line 356-357).” The observation that the reported mutation E228K has substantially higher occupancy of the active state may be because the addition of positive charge of residue K at the calcium binding site mimics the stabilization effect of calcium binding by electrostatic interaction.

Fig 5 b and C labels are switched

Line 144-144 Fig 1c did not describe the construct

Reviewer #2 (Remarks to the Author):

1. The authors have addressed all of my previous concerns.

2. Recently, another paper on the cryo-EM structures of CaSR has been published (Park et al., PNAS 118, e2115849118, 2021). The authors could cite this paper along with the others.

Reviewer #3 (Remarks to the Author):

The authors have added significant amount of data and technical details to address all my questions, as well as those of the other reviewers. As a result, the manuscript looks much clearer now and I believe that it meets the standard to be published in Nature Communications.

REVIEWERS' COMMENTS

Reviewer #1 (Remarks to the Author):

In this revision, Schamber and Vafabakhsh have responded to the majority of the previously concerns with additional data and revision. The following additional revisions are suggested:

1). Major conclusions are based on the assumption that smFRET constructs accurately reflect the cellular CaSR in live cell activation processes. The smFRET signals were achieved using SNAP-tag with labeling and dyes in a fixed cell lysate measurement, or were based on non-natural amino acid UAA substitution at either position D451 (D451UAA0), or at E593 (E593UAA) with dye labeling. Please add dose response curves for these constructs and compare with w.t without labeling, using live cells with calcium imaging, to fully justify that “these constructs are active with EC50 comparable to wildtype receptor” (line 146-148), since CaSR’s EC50 changes and can depend on experimental buffer conditions.

We had the suitable plasmid for the calcium imaging for E593UAA construct and were able to perform those experiments. We have now included the Ca²⁺ titration data for E593UAA as the new Supplementary Fig. 3b.

2) Please modify the following sentence: “Finally, while none of the structures show Ca²⁺ binding at the negative patch on LB2 (Supplementary Fig. 5d), it is possible that transient or weak binding of Ca²⁺ at this interface can contribute to CaSR activation by screening the negative charges, as suggested in a previous study (Huang, et al., Biochemistry., 2009, & Zhang, et al., 2016, Science Advances) Line 433-434)”. A calcium binding site was reportedly involved in E228 and E229 at the negatively charged surface between lobe 2 (LB2) by Huang et al. (Biochem 2007), based on purified protein fragments, mutational studies (E228I, E229I), and calcium response functional assays.

We understand the reviewer's point. However, the quoted sentence from our revised manuscript that “none of the structures show Ca²⁺ binding at the negative patch on LB2 (Supplementary Fig. 5d)...” is simply based on analyzing all the published structures (Supplementary Fig. 5d). No atomic structure that is available on the PDB to date show Ca²⁺ ion in the lower lobe interface or coordinated by E228 or E229. In our own experience and based on the available literature in the GPCR field, mutagenesis and fragment analysis results can be confounded by indirect and allosteric effects and don't “prove” the location of a binding site but rather perhaps support the inference. We have acknowledged this possibility that a weak and transient binding that is not captured in the structures could play a role in activation of CaSR (Lines 452-456). We have now further modified this sentence to even stronger reflect the interpretation from the Biochem 2007 paper and we have now included this reference.

“Finally, while none of the atomic structures show Ca²⁺ binding at the negative patch on LB2 (Supplementary Fig. 5d), it is possible that transient or weak binding of Ca²⁺ at this region could contribute to CaSR activation via screening the negative charges. This interpretation is consistent with previous reports based on mutagenesis and fragment analysis 13,60,61 , and positively charged residues could emulate this proposed effect.”

In previously reported studies by Huang et al. 2007 and Zhang et al, 2016, removal of calcium binding ligand residues E228I and/or E229I significantly increased the EC50 of intracellular calcium responses of this receptor, supporting the role of calcium binding at this negatively charged patch location in the activation of the dimer form. These previously reported results are different from the statement in this manuscript that “the regulatory effect of negatively charged LB2 patch is independent of Ca²⁺” (line 356-357).” The observation that the reported mutation E228K has substantially higher occupancy of the active state may be because the addition of positive charge of residue K at the calcium binding site mimics the stabilization effect of calcium binding by electrostatic interaction.

We have now acknowledged this speculative possibility in the discussion (Line 456).

Fig 5 b and C labels are switched

We thank the reviewer for pointing out this error. we have corrected the figure caption.

Line 144-144 Fig 1c did not describe the construct

This figure is not describing a specific construct, rather it is a generic schematic for smFRET experiments.

Reviewer #2 (Remarks to the Author):

1. The authors have addressed all of my previous concerns.
2. Recently, another paper on the cryo-EM structures of CaSR has been published (Park et al., PNAS 118, e2115849118, 2021). The authors could cite this paper along with the others.

We thank the reviewer for bringing this paper to our attention. We have now added this citation along with the others. (Lines 81-83).

Reviewer #3 (Remarks to the Author):

The authors have added significant amount of data and technical details to address all my questions, as well as those of the other reviewers. As a result, the manuscript looks much clearer now and I believe that it meets the standard to be published in Nature Communications.

We thank the reviewer for their constructive feedback during the revision process.